# circFL-seq reveals full-length circular RNAs with rolling circular reverse transcription and nanopore sequencing

Zelin Liu[1], Changyu Tao[2], Shiwei Li[3], Minghao Du[4], Yongtai Bai[3], Xueyan Hu[1], Yu Li[5], Jian Chen[5], Ence Yang[1,4,5]*

[1]Institute of Systems Biomedicine, Department of Medical Bioinformatics, School of Basic Medical Sciences, Peking University Health Science Center, Key Laboratory for Neuroscience, Ministry of Education/National Health Commission of China , NHC Key Laboratory of Medical Immunology (Peking University), Beijing, China; [2]Department of Human Anatomy, Histology & Embryology, School of Basic Medical Sciences, Peking University Health Science Center, Beijing, China; [3]Department of Radiation Medicine, School of Basic Medical Sciences, Peking University Health Science Center, Beijing, China; [4]Department of Microbiology & Infectious Disease Center, School of Basic Medical Science Peking University Health Science Center, Beijing, China; [5]Chinese Institute for Brain Research, Beijing, China

**\*For correspondence:**
yangence@pku.edu.cn

**Competing interest:** The authors declare that no competing interests exist.

**Abstract** Circular RNAs (circRNAs) act through multiple mechanisms via their sequence features to fine-tune gene expression networks. Due to overlapping sequences with linear cognates, identifying internal sequences of circRNAs remains a challenge, which hinders a comprehensive understanding of circRNA functions and mechanisms. Here, based on rolling circular reverse transcription and nanopore sequencing, we developed circFL-seq, a full-length circRNA sequencing method, to profile circRNA at the isoform level. With a customized computational pipeline to directly identify full-length sequences from rolling circular reads, we reconstructed 77,606 high-quality circRNAs from seven human cell lines and two human tissues. circFL-seq benefits from rolling circles and long-read sequencing, and the results showed more than tenfold enrichment of circRNA reads and advantages for both detection and quantification at the isoform level compared to those for short-read RNA sequencing. The concordance of the RT-qPCR and circFL-seq results for the identification of differential alternative splicing suggested wide application prospects for functional studies of internal variants in circRNAs. Moreover, the detection of fusion circRNAs at the omics scale may further expand the application of circFL-seq. Taken together, the accurate identification and quantification of full-length circRNAs make circFL-seq a potential tool for large-scale screening of functional circRNAs.

## Introduction

Circular RNAs (circRNAs), a class of covalently closed RNA molecules formed via back-splicing (BS) or lariat precursors, are involved in various biological processes and pathogenesis by fine-tuning the eukaryotic gene regulatory network (*Kristensen et al., 2019*; *Chen, 2020*). CircRNAs directly or indirectly regulate target gene expression via diverse mechanisms, which are largely determined by circRNA sequence features. For example, circRNAs may complementarily bind to miRNAs as sponges (*Hansen et al., 2013*; *Memczak et al., 2013*); circRNAs may interact with proteins as scaffolds (*Du et al., 2017*) or structural components (*Li et al., 2015*) based on sequence motifs; and several circRNAs are also able to translate into peptides (*Zhang et al., 2018*; *Gao et al., 2021* ) through

internal ribosome entry sites. Thus, full-length sequences of circRNAs have become the foundation for ascertaining their biological functions in transcriptional plasticity and complexity.

By detecting the back-splicing junctions (BSJs) of circRNAs with deep sequencing, short-read RNA sequencing discriminates circRNAs with low expression (as low as 1% polyadenylated RNA; *Salzman et al., 2013*) from their linear cognates. The full-length sequences of short circRNAs (<500 nt) can be inferred from a patchwork of BSJs and short fragments via bioinformatic approaches (*Zheng et al., 2019*; *Wu et al., 2019b*). However, a full understanding of circRNA isoforms is impossible by using short reads. Single-molecule long-read sequencing has shown methodological advances in identifying circRNAs at the isoform level. Pacific Biosciences (PacBio) sequencing has been applied to PCR products for target full-length circRNA sequences in a low-throughput and high-cost way (*You et al., 2015*). Very recently, several Oxford Nanopore Technology (ONT)-based methods have also been employed in genome-wide full-length circRNA reconstruction (*Rahimi et al., 2019*; *Xin et al., 2021a*; *Zhang et al., 2021a*).

Here, we developed a high-throughput circRNA sequencing method, termed circFL-seq, with rolling circular reverse transcription (RCRT) (*Boss and Arenz, 2020*; *Das et al., 2019*) and nanopore long-read sequencing for the identification of circRNA at the isoform level. With a customized computational pipeline, we identified 77,606 high-quality full-length circRNA isoforms from seven cell lines and two human tissues. We validated circFL-seq for full-length circRNA detection and quantification by comparison to annotated circRNAs, RNA-seq, isoCirc, CIRI-long, and RT-qPCR results. By providing full-length circRNA sequences, circFL-seq allowed the study of sequence features, alternative splicing

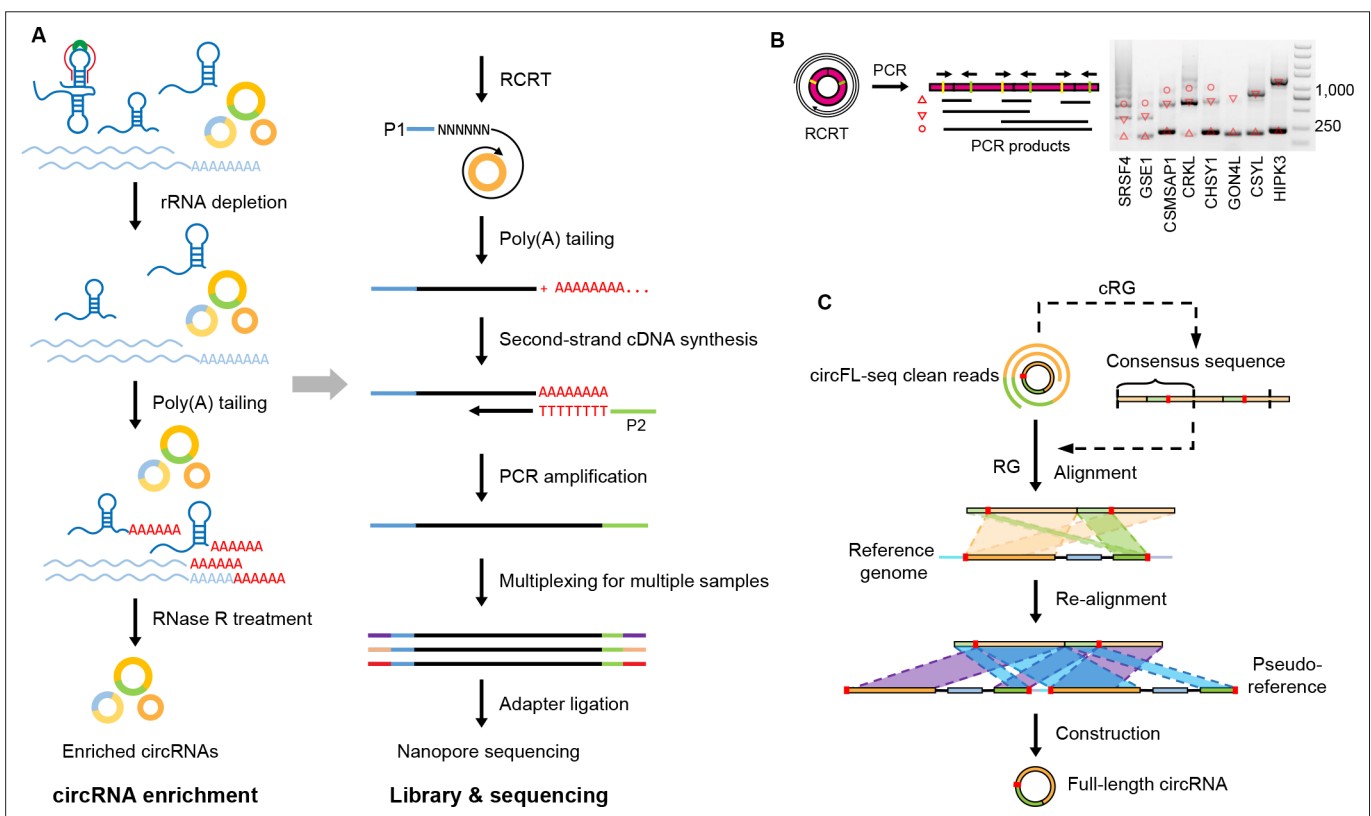

**Figure 1.** Diagram of circFL-seq workflow. (**A**) Experimental operation of circFL-seq consisted of circRNA enrichment, library construction, and nanopore sequencing. (**B**) PCR validation of rolling circle products from the circFL-seq cDNA library. The yellow and green lines indicate the positions of the PCR primers. The upward triangle, downward triangle, and circle symbols denote the 0-circle, 1-circle, and 2-circle cDNA products. (**C**) Computational pipeline of circFL-seq. circFL-seq clean reads were directly used in RG mode or were self-corrected for consensus sequences in cRG mode to reconstruct full-length circRNAs. circRNA, circular RNA.

The online version of this article includes the following figure supplement(s) for figure 1:

**Source data 1.** Original figures of gels.

**Figure supplement 1.** Sanger sequencing of rolling circular bands.

(AS), and differential expression at the isoform level. In addition, circFL-seq showed the ability to detect fusion circRNAs (f-circRNAs), which were further experimentally validated. Taken together, circFL-seq benefits from RCRT and long-read sequencing and has shown advantages for identifying high-quality full-length circRNAs at a low cost.

## Results

### Sequencing full-length circRNA with circFL-seq

We developed the circFL-seq approach that utilized RCRT with long-read sequencing for full-length circRNA profiling (*Figure 1A*). First, circRNA was enriched by rRNA depletion, poly(A) tailing (to increase the efficiency of RNase R *Xiao and Wilusz, 2019*), and RNase R treatment. Then, first-strand cDNA was synthesized with random primers (P1-N$_6$) by RCRT for circRNA and regular reverse transcription (RT) for linear residuals. After tailing poly(A) of the first strand and synthesizing the second strand with an anchor primer (P2-T$_{24}$), the double-strand cDNA was amplified with P1 and P2 to construct the final circFL-seq library. To benchmark full-length circRNA sequences of the circFL-seq library, eight known circRNAs were selected for PCR validation in the HEK293T library. Rolling circles of target circRNA (*Figure 1B*; *Figure 1—source data 1*) and the full-length sequences obtained with Sanger sequencing (*Figure 1—figure supplement 1*) indicated that the circRNAs were successfully amplified. Finally, the circFL-seq library was sequenced by long-read sequencing on the ONT platform.

Based on the nature of RCRT products in circFL-seq libraries, full-length sequences of circRNAs were reconstructed and structurally annotated with a customized computational pipeline. For the reference guide (RG) mode, clean reads (qscore ≥7) were aligned to the reference genome to identify potential back-spliced junctions (BSJs) by the sign of chiastic overlapping segments (*Figure 1C*). Then, reads were realigned to pseudo-references, generated by concatenating two sequences of potential BSJ regions, to accurately localize circRNA BSJs. In addition, full-length structures were adjusted with multiple alignments from rolling circular segments to improve the construction. To complement the RG mode for low-quality reads, the cRG mode identified consensus sequences (CS) from clean reads with two or more rolling circles, and triply duplicated CS were used as query sequences in RG mode to improve circRNA detection. The strand origin of circRNAs, especially non-canonical BSJs, was predicted by the primer sequences P1, P2, and P2-T$_{24}$.

### CircRNA profiling of eight libraries

To assess the performance of circFL-seq, we applied circFL-seq to eight libraries of six human cell lines (two replicates from HeLa cells, two replicates from SKOV3 cells, one from MCF7 cells, one from VCaP cells, one from SH-SY5Y cells, and one from HEK293T cells) and produced 30 M clean reads sequenced by using one PromethION Flow Cell (*Figure 2—figure supplement 1A*). We detected 197,252 isoforms of 162,409 circRNA BSJs from 1.3 M reads that contained full-length circRNA sequences (*Figure 2A*; *Figure 2—figure supplement 1B*; *Figure 2—figure supplement 2A*; *Supplementary file 1*), 2% of which were additionally detected by cRG mode, which refined low-quality reads with CS (*Figure 2—figure supplement 2B-D*). Quantification of replicate samples at both the BSJ and isoform levels was highly consistent (Pearson's r>0.93; *Figure 2A and B*; *Figure 2—figure supplement 3*).

Based on BSJs and boundary exons identified by circFL-seq, we classified circRNAs into seven types: exonic, intronic, novel splicing site (NSS), intergenic, novel UTR, antisense, and read-through (*Figure 2—figure supplement 4*). For high-quality circRNA isoforms (read counts≥5), the exonic type accounted for 73.2% of the identified circRNA species, while only 0.1% were from read-through (*Figure 2C*). Most exonic types (99.8%) could be verified in at least one database (*Glažar et al., 2014*; *Chen et al., 2016*; *Dong et al., 2018*; *Liu et al., 2019*; *Vo et al., 2019*; *Huang et al., 2021*; *Figure 2C*), which could be attributed to their higher expression levels (*Figure 2—figure supplement 5*). The median length of each type varied from 402 nt of the intergenic type to 618 nt of read-throughs (*Figure 2D*), indicating that a notable proportion of circRNAs>500 nt cannot be reconstructed by RNA-seq. Read-through circRNAs contained more exons (*Figure 2E*), while the exon length of single-exon circRNA was significantly longer than that of multiple-exon circRNA (p<2.2×10$^{-16}$; *Figure 2F*).

From 65,656 isoforms of 35,251 high-quality circRNA BSJs (read counts≥5), we identified 23,267 internal AS events (*Supplementary file 2*), including 44.3% exon skipping (ES), 28.1% alternative 3′

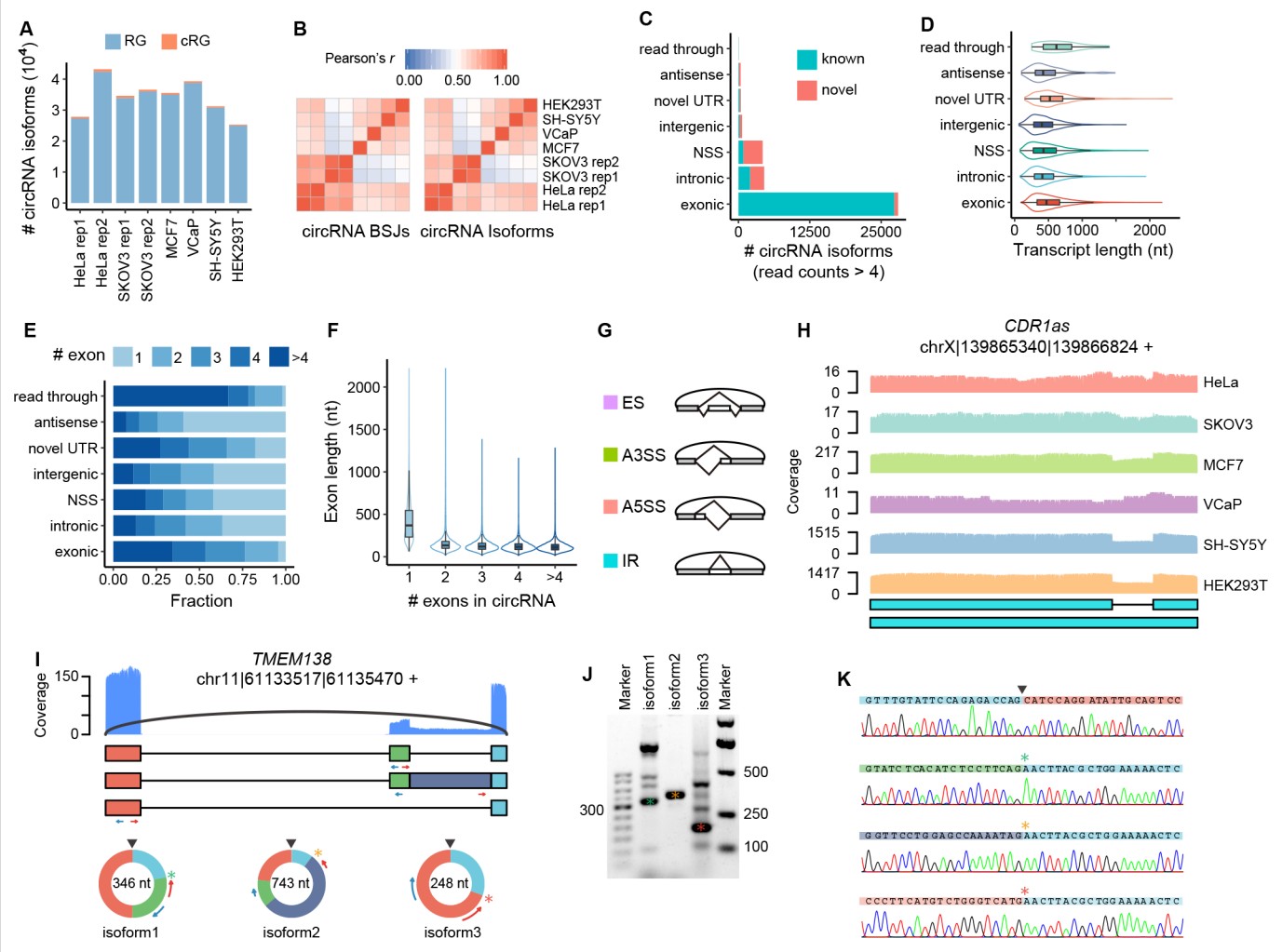

**Figure 2.** Analysis of full-length circRNA in eight samples. (**A**) Stacked bar plot represents the number of full-length circRNA isoforms detected by RG and cRG for six cell lines. (**B**) Expression correlation matrix for circRNA BSJs and isoforms of all samples. The color scale corresponds to Pearson's correlation coefficient. (**C**) Stacked bar plot represents the number of circRNA isoforms with read counts≥5 from known or novel BSJs based on the circRNA database. (**D**) Boxplot showing the length distribution per isoform for circRNA isoforms with read counts≥5 in all samples. Box lefts or rights are lower or upper quartiles, the bar is the median, and whiskers are the median ±1.5×interquartile range. (**E**) Stacked bar plot showing the fraction of exon numbers per isoform for circRNA isoforms with read counts≥5 in all samples. (**F**) Boxplot showing the length distribution per exon for circRNA isoforms with read counts≥5 in all samples. Box bottoms or tops are lower or upper quartiles, the bar is the median, and the whiskers are the median ±1.5×interquartile range. (**G**) Diagram of four types of alternative splicing (AS) events in circRNA: exon skipping (ES), alternative 3' splice site (A3SS), alternative 5' splice site (A5SS), and intron retention (IR). (**H**) Plot showing the coverage of full-length circRNA reads in the position of *CDR1as* for circFL-seq data of six cell lines (replicate data were merged). Structures of the two isoforms of *CDR1as* are shown at the bottom. (**I–K**) AS events (one ES and one IR) of circ-*TMEM138* detected by circFL-seq (**I**), agarose gel electrophoresis (**J**), and Sanger sequencing (**K**). Red/blue arcs are forward/reverse primers for validation of back-splicing junctions (BSJs) and forward splicing junctions (FSJs). Asterisks denote FSJs. Downward triangles denote BSJs. BSJ, back-splicing junction; circRNA, circular RNA; RG, reference guide.

The online version of this article includes the following figure supplement(s) for figure 2:

**Figure supplement 1.** Clean read distribution of circFL-seq data of six cell lines.

**Figure supplement 2.** CircRNA reads identified from circFL-seq data of six cell lines.

**Figure supplement 3.** Scatter plot showing the correlation of circRNA at the BSJ level (**A, B**) and isoform level (**C, D**) between circFL-seq replicates.

**Figure supplement 4.** Diagram of circRNA types.

**Figure supplement 5.** Cumulative distribution of read counts for circRNA isoforms identified by circFL-seq from six cell lines.

**Figure supplement 6.** CircRNAs with exon skipping validated by RT-PCR and Sanger sequencing in HeLa cells.

**Figure supplement 7.** CircRNAs with alternative 3'/5' splicing sites (A3SS for circRNAs from *MCU* and *MRS2*, A5SS for circRNA from *SNX25*) were

*Figure 2 continued on next page*

*Figure 2 continued*

validated by RT-PCR and Sanger sequencing in HeLa cells.

**Figure supplement 8.** CircRNAs with intron retention validated by RT-PCR and Sanger sequencing in HeLa cells.

splicing site (A3SS), 24.1% alternative 5′ splicing site (A5SS), and 3.5% intron retention (IR) events (*Figure 2G*). Specifically, IR events increased circRNA length by 289 nt on average (from 363 to 652 nt), which may influence their cellular localization (*Huang et al., 2018*). circFL-seq accurately detected the full length of two isoforms (1485 and 1301 nt) generated by the IR event of *CDR1as* (*Hansen et al., 2011*; *Figure 2H*). We further experimentally verified 13 AS events (5 of ES, 3 of A3SS, 1 of A5SS, and 4 of IR) of 23 isoforms from 11 high-quality circRNAs in the HeLa cell line. With divergent primers to amplify both BSJs and alternative spliced forward splicing junctions (FSJs), all but an A3SS event were validated by Sanger sequencing (*Figure 2I–K*; *Figure 2—figure supplements 6–8*).

## Comparison with RNA-seq, isoCirc, and CIRI-long for circRNA detection

Based on read-spanning BSJs, short-read sequencing has long been used for genome-wide characterization of circRNAs. Thus, for comparison, eight RNA-seq libraries (150 bp ×2) with the same circRNA enrichment method for the same six cell lines (Pearson's r>0.90 for replicates, *Figure 3—figure supplement 1*) were generated and sequenced by the Illumina HiSeq X Ten platform. A total of 95,371 circRNA BSJs were detected by CIRI2 (*Gao et al., 2018*; *Supplementary file 3*). BSJ reads accounted for 0.15–0.32% of all RNA-seq reads, an amount 10 times lower than that of full-length circRNA reads with circFL-seq (2.2–8.5%). Compared to known BSJs that have been annotated in the database, the proportion of overlapping BSJs identified by RNA-seq seemed to be larger than that identified by circFL-seq (76.7% vs. 40.3%) (*Figure 3—figure supplement 2*). However, when focusing on high-quality BSJs (read counts≥5), the proportion obtained with circFL-seq (78.9%) was similar to that obtained with RNA-seq, indicating that more reads are required to confidently identify circRNAs by ONT (*Figure 3—figure supplement 3*). As expected, the full-length construction by RNA-seq was highly dependent on circRNA length (*Figure 3—figure supplement 4*). Approximately 96.3% of full-length circRNAs reconstructed by RNA-seq were less than 500 nt in length, while 44.4% of circRNA isoforms detected by circFL-seq were more than 500 nt in length.

A recent method, isoCirc, identified full-length circRNAs by rolling circle amplification (RCA) followed by nanopore sequencing. With a single circFL-seq library of the HEK293 cell line sequenced by one MinION Flow Cell, we compared the features of the RCRT strategy and RCA strategy. circFL-seq produced more full-length circRNA reads per $10^9$ raw bases than isoCirc (11,791 vs. 5820) (*Figure 3—figure supplement 5A-C*) but identified fewer circRNA isoforms (*Figure 3—figure supplement 5D*). However, 75.3% of circFL-seq-detected BSJs were annotated in the database, while the percentage of isoCirc was 43.8% (*Supplementary file 4*). The difference might be caused by the higher read coverage for confident circRNAs of circFL-seq (*Figure 3—figure supplement 5E*) or by the higher sensitivity of circRNA detection of isoCirc. Although the ground truth of all BSJs was unknown, we found that circRNAs with higher read counts were more likely to be included in the database for both methods (*Figure 3—figure supplement 5F-I*). Thus, we evaluated the precision of the two methods by top expressed circRNAs, which could eliminate the effect of the high noise of nanopore reads and the disturbance of low-expressed circRNAs without annotation in the database. All the top 100 expressed BSJs identified by circFL-seq and RNA-seq (collected from isoCirc study) were annotated in databases. In contrast, 22 of the top 100 BSJs identified by isoCirc were not supported in the databases or the RNA-seq and circFL-seq results (*Figure 3—figure supplement 5M*). Specifically, two products among the 22 BSJs were from histone genes with linear RNAs resistant to RNase R degradation (*Xiao and Wilusz, 2019*). We also failed to experimentally validate the two potential circular products with divergent primers. At the isoform level in the HEK293 cell line, 87.0% of the common BSJs (10,702/12,307) between circFL-seq and isoCirc identified at least one identical isoform. Inconsistent isoforms usually had lower read counts with both circFL-seq and isoCirc (*Figure 3—figure supplement 5N,O*), suggesting that accurate construction of full-length circRNA requires higher read coverage. In addition, circFL-seq identified more than two times ES, A3SS, and A5SS but similar numbers of IR events to isoCirc (*Supplementary file 4*), which may be due to the more full-length reads of circFL-seq.

We then compared circFL-seq to isoCirc in two normal human tissues (brain and testis) with high sequencing depth. isoCirc built multiple libraries for each sample and sequenced on multiple MinION Flow Cells to obtain a high sequencing depth, while circFL-seq built one library for each tissue and sequenced the two libraries on one PromethION Flow Cell to obtain a comparable sequencing depth. isoCirc and circFL-seq detected 79,312 and 34,046 known BSJs, respectively. Known BSJs detected by isoCirc were more likely (38,846 vs. 2511) to be expressed at low levels (read counts=1), indicating that more libraries were more sensitive for detecting circRNAs, overcoming bias and artifacts related to sampling, RNase R treatment, cDNA synthesis, and amplification.

Both circFL-seq and CIRI-long employed an RCRT strategy for library construction. Although the two methods were applied in different species (human vs. mouse), the computational pipelines were compatible and could be compared with each other. We employed the computational pipeline of circFL-seq and CIRI-long to analyze HEK293 cells library of circFL-seq and a mouse brain library of CIRI-long, respectively. circFL-seq detected 27,869 BSJs from HEK293 cells ( 75.3% known in the database) and 18,396 BSJs from the mouse brain ( 68.6% known in the database), while CIRI-long detected 15,242 BSJs from HEK293 cells ( 76.9% known in the database) and 9258 BSJs from the mouse brain ( 69.9% known in the database) (*Supplementary file 5*). Thus, the computational pipeline of circFL-seq was more sensitive with similar precision for BSJ detection. For AS events, circFL-seq identified more than 4.5 times the ES events, 12.4 times the A3SS events, 8.5 times the A5SS events, and 1.5 times the IR events, suggesting a better sensitivity of circFL-seq in AS event detection, although ground-truth circRNA isoforms were required for performance evaluation (*Supplementary file 5*).

## Evaluation of quantification of full-length circRNAs

With the benefit of high read coverage, circFL-seq is able to quantify circRNA expression levels. For BSJs identified by circFL-seq and RNA-seq, the amounts between the two methods were significantly correlated (Pearson's r=0.41–0.68, *Figure 3A*; *Figure 3—figure supplement 6*). We then detected differentially expressed circRNAs (DECs) between HeLa and SKOV3 cell lines as a test. For the highly expressed BSJs (read counts≥10 in at least two samples and detected by both methods), the fold changes in HeLa to SKOV3 cells were highly concordant with the RNA-seq results (Pearson's r=0.78, *Figure 3B*). By using DESeq2 (*Love et al., 2014*), circFL-seq detected 89 DECs, 58 of which were also identified by RNA-seq. We next selected 16 circRNAs with a wide range of fold changes (7 downregulated, 5 stable, and 4 upregulated in the HeLa cell line) to validate the DECs via RT-qPCR. The consistent circFL-seq and RT-qPCR results for both total RNA (*Figure 3—figure supplement 7A-C*) and RNase R-treated (*Figure 3C and D*) samples further supported the capabilities of BSJ quantification.

We next evaluated the quantification performance of circFL-seq for circRNA AS. From 87 BSJs that had at least two isoforms detected by both HeLa and SKOV3 cell lines, 193 isoforms were used to quantify the internal variants by the ratio of transcripts (target isoforms to total isoforms from the same BSJ). A total of 90 transcripts had more than 0.1 ratio differences between HeLa and SKOV3 cell lines. For example, circ-*PLOD2* showed a higher ratio of transcripts without ES in the HeLa cell line (0.83 to 0.45) (*Figure 3E*). Although RNA-seq data also detected differential read coverage at the skipped exon of *PLOD2*, short reads were unable to discriminate different sources of ES events, that is, linear or circular isoforms. We selected 18 isoforms of 9 circRNA BSJs from a wide range of ratio differences to perform experimental validation. Both the transcript ratios and ratio differences were consistent with the RT-qPCR results (*Figure 3F and G*; *Figure 3—figure supplement 7D-F*), supporting the advantage of circFL-seq for quantification at the isoform level.

## circFL-seq reveals fusion circRNAs with long-read sequencing

F-circRNA is a specific type of circRNA containing two fusion junctions (one junction may come from gene fusion and the other may be produced by BS) and has been found to play important roles in cancer (*Guarnerio et al., 2016*). Although short reads of RNA-seq may detect both fusion junctions separately, they usually fail to detect f-circRNA for the short read that is not able to combine two fusions in one fragment. circFL-seq was able to detect f-circRNA with rolling circular reads separately mapped to two loci in different chromosomes or a >1 Mbp distance in the same chromosome (*Figure 4A*). With circFL-seq data, we identified six high-quality f-circRNA isoforms (read count ≥5) affiliated with two fusion genes in the MCF7 cell line. For the five isoforms fused by *GBF1* and *MACROD2* (*Figure 4B*), we selected two major isoforms (circ_290 nt for 35% and circ_408 nt for 26%) to perform full-length

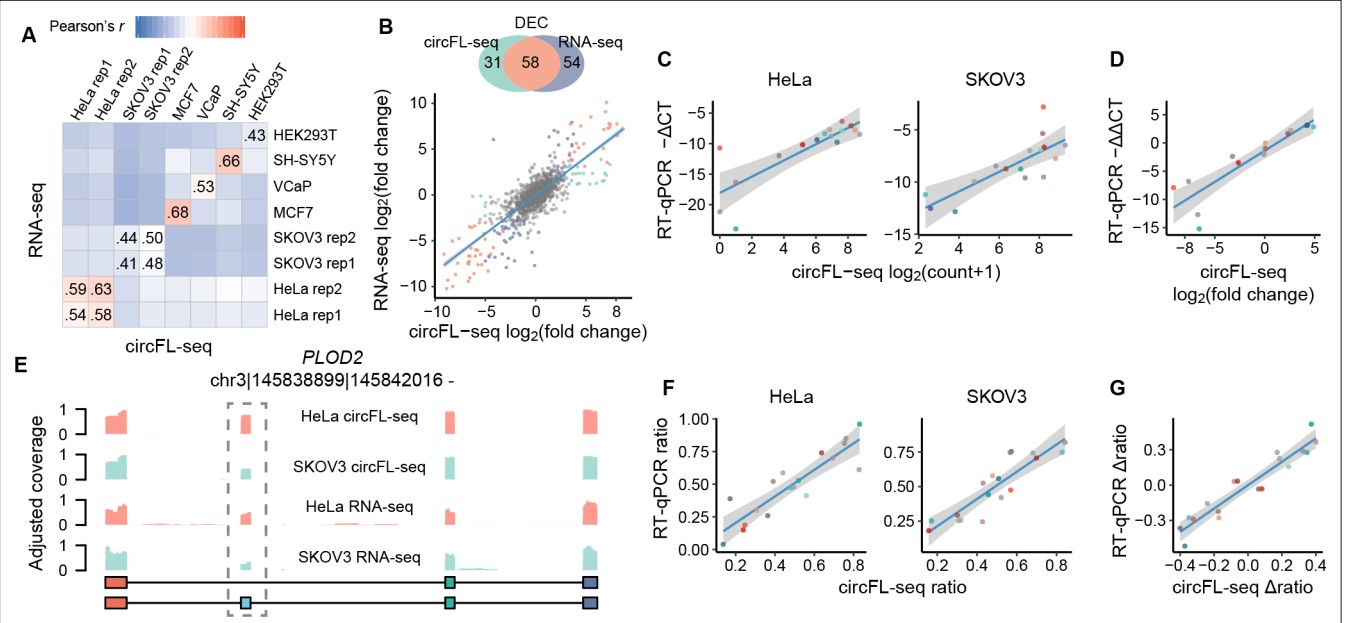

**Figure 3.** Quantification of circRNA at the BSJ and isoform levels. (**A**) Expression correlation matrix of circRNA BSJ quantified by circFL-seq and RNA-seq for six cell lines. The numbers in the matrix represent Pearson's correlation coefficients. (**B**) Comparison of differentially expressed circRNA (DEC) detection between circFL-seq and RNA-seq. Top panel: Venn diagram showing the number of DECs detected by circFL-seq (green), RNA-seq (purple), and both methods (orange). Bottom panel: scatter plot showing the correlation of fold change (log base 2) for HeLa and SKOV3 cells between circFL-seq and RNA-seq. (**C**) Scatter plot showing the correlation of the expression levels of 16 circRNA BSJs for HeLa (left) and SKOV3 (right) cells between circFL-seq and RT-qPCR. (**D**) Scatter plot showing the correlation of fold changes (log base 2) of the 16 BSJs for HeLa and SKOV3 cells between circFL-seq and RT-qPCR. (**E**) Plot showing the adjusted coverage of full-length circRNA reads and RNA-seq reads in the position of circRNA from *PLOD2*. The circular structures of the two circRNA isoforms are shown in the lower panel. (**F**) Scatter plot showing the correlation of the transcript ratio of 18 circRNA isoforms from nine circRNA BSJs (each BSJ has two isoforms) for HeLa (left) and SKOV3 (right) cells between circFL-seq and RT-qPCR. The relative expression of target BSJs/isoforms quantified by RT-qPCR was determined with RNase R-treated samples and *GAPDH* from total RNA without RNase R treatment as a reference. (**G**) Scatter plot showing the correlation of the differential ratio (Δratio) of the 18 isoforms for HeLa and SKOV3 cells between circFL-seq and RT-qPCR. The shaded areas denote 95 % confidence intervals. BSJ, back-splicing junction; circRNA, circular RNA.

The online version of this article includes the following figure supplement(s) for figure 3:

**Figure supplement 1.** Correlations of circRNA BSJs among RNA-seq samples from six cell lines.

**Figure supplement 2.** Venn diagram of BSJs detected by circFL-seq, RNA-seq, and database.

**Figure supplement 3.** CircRNA read distribution of eight samples of six cell lines.

**Figure supplement 4.** Comparison of circFL-seq and RNA-seq for length of full-length circRNA of six cell lines.

**Figure supplement 5.** Comparison of circFL-seq and isoCirc for full-length circRNA detection in the HEK293 cell line.

**Figure supplement 6.** Scatter plot showing the correlation of circRNA BSJs between circFL-seq and RNA-seq samples of six cell lines.

**Figure supplement 7.** Evaluation of circRNA quantification between circFL-seq and RT-qPCR.

validation. As expected, more than one product was observed (*Figure 4C*), and the rolling circles of full-length sequences were validated by Sanger sequencing (*Figure 4D*, *Figure 4—figure supplement 1*). For comparison, we also used linear RNA from the MCF7 cell line and RNase R-treated RNA from the HeLa cell line as templates. Intriguingly, we validated the fusion junction of E2–E4 with linear RNA as a template (*Figure 4C*), suggesting the existence of linear RNA production from the fusion gene. The decreased expression of junctions E2–E4 after RNase R treatment also supported the linear production, which further suggested that the fusion direction is *GBF1* to *MACROD2* (*Figure 4E*). In addition, the f-circRNA fused by two antisense genes (*PRICKLE2-AS1* and *PTPRT-AS1*) was experimentally validated (*Figure 4F*; *Figure 4—figure supplement 2*). All the major fusion junctions of *GBF1/MACROD2* and E3-E1 of *PRICKLE2-AS1/PTPRT-AS1* were validated by RNA-seq of the MCF7 cell line (*Figure 4G*).

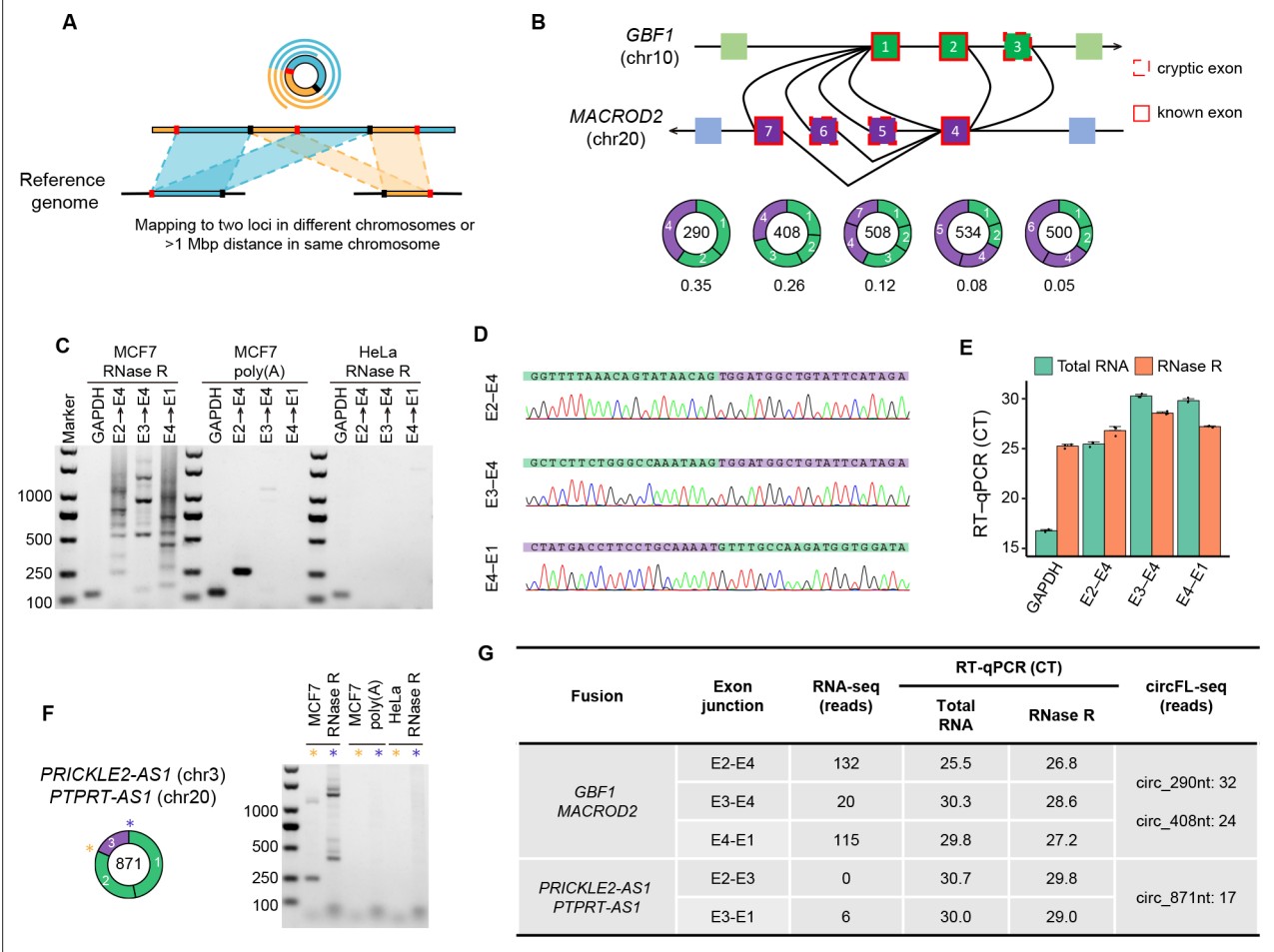

**Figure 4.** Detection and validation of fusion circRNA (f-circRNA) in the MCF7 cell line. (**A**) Diagram of identification of f-circRNA with circFL-seq data. (**B**) Diagram of five high-quality f-circRNA isoforms (read counts≥5) fused by *GBF1* and *MACROD2*. The transcript ratio represents the fractions of the isoforms. (**C–E**) Validation of f-circRNA junctions from *GBF1/MACROD2* by agarose gel electrophoresis (**C**), Sanger sequencing (**D**), and RT-qPCR (**E**). (**C**) Agarose gel electrophoresis showing the RT-PCR products of f-circRNA junctions with RNase R-treated MCF7 and HeLa RNA and poly(A) selected MCF7 RNA as a template. (**F**) Agarose gel electrophoresis showing the RT-PCR products of f-circRNA junctions from *PRICKLE2-AS1/PTPRT-AS1*. (**G**) Information on five f-circRNA junctions detected by circFL-seq, RNA-seq, and RT-qPCR.

The online version of this article includes the following figure supplement(s) for figure 4:

**Figure supplement 1.** Sanger validation of sequences of f-circRNA from *GBF1/MACROD2*.

**Figure supplement 2.** Sanger validation of the sequence of f-circRNA from *PRICKLE2-AS1/PTPRT-AS1*.

# Discussion

In this study, we established a full-length circRNA sequencing method, circFL-seq, by nanopore sequencing. Compared to short-read sequencing, which is limited to reconstructing short circRNAs (<500 nt), long-read sequencing with circFL-seq has shown advances in comprehensively identifying full-length circRNAs of all sizes (64–2334 nt in our data and more than 40% identified isoforms >500 nt). Very recently, isoCirc (**Xin et al., 2021a**) and CIRI-long (**Zhang et al., 2021a**) also employed nanopore sequencing to identify full-length circRNAs by utilizing rolling circles. circFL-seq and CIRI-long employ the RCRT strategy, while isoCirc employs the RCA strategy. Benefiting from RCA with a circular cDNA template from ligation of the RT product, isoCirc produces longer reads (up to 50 kb) containing more rolling circles than RCRT-based methods (ca. 1 kb) and thus has an advantage in error correction during CS identification. As a trade-off, the cost of each full-length read is much higher in isoCirc than in circFL-seq and CIRI-long. In addition, the unique ligation of cDNA in isoCirc may introduce false positives for circRNA detection by ligating cDNA from residual linear RNA or truncated cDNA of circRNA, which is hardly totally recognized and removed by computational analysis.

In contrast to isoCirc, both the circFL-seq and CIRI-long-based methods employ RCRT to produce rolling circles during first-strand synthesis. However, circFL-seq synthesizes second-strand cDNA with an anchor primer, while CIRI-long uses template switching. Thus, these two methods are more resistant to the interference of residual linear RNA and generate shorter full-length reads of circRNA than isoCirc. Although decreasing the sequencing cost of nanopore, shorter full-length reads have to perform error correction among reads in addition to intra reads. Thus, RCRT-based methods are not as sensitive as isoCirc with the same number of full-length reads. The computational pipelines of circFL-seq and CIRI-long are compatible given their very similar experimental protocols. Both of these methods calculate CS from tandem repeats of reads to refine low-quality sequences, followed by localization of CS. However, CS may ignore reads that are less than two full circles. In contrast, the RG mode of circFL-seq identified 56% more full-length circRNAs with fewer than two circles in our data, which helps to improve the sensitivity of circRNA detection. In addition, circFL-seq leverages primer sequences to adjust the strand origin of circRNA, especially for BSJ with a non-canonical splicing motif. For details, we summarize the three methods in *Supplementary file 6*.

Previous short-read RNA sequencing was used to quantify circRNAs based on BSJ-spanning reads, while circFL-seq with full-length reads was employed to finely quantify circRNAs at the isoform level. Thus, circFL-seq is able to identify differential internal AS events, helping to unravel the function of sequence variants. In our comparisons, circFL-seq was more sensitive than isoCirc and CIRI-long for AS detection, although the performance required full evaluation by ground-truth circRNA isoforms. Another novel application of full-length circRNA sequencing is to identify f-circRNA, which has been found to play important roles in cancer pathogenesis (*Guarnerio et al., 2016*) and has become a promising biomarker for liquid biopsy (*Tan et al., 2018a*). Current methods (*Guarnerio et al., 2016*; *Tan et al., 2018a*; *Tan et al., 2018b*; *Wu et al., 2019a*) validated the proposed f-circRNA by designing divergent primers on known gene fusion junctions and thus were inefficient and possibly ignored f-circRNA from unknown gene fusions. With cancer cell line data, circFL-seq showed its ability to high-throughput identify f-circRNA, which will promote our knowledge of f-circRNA.

Overall, circFL-seq is able to identify and quantify circRNAs at the isoform level and is more sensitive in circRNA AS event detection and f-circRNA detection. In addition, circFL-seq has been applied in a large-scale sequencer, PromethION, with pooling multiple libraries, which will further decrease the sequencing cost to twice that of RNA-seq. Thus, our developed circFL-seq process is an effective and affordable high-throughput full-length circRNA sequencing method for screening functional circRNAs at the omics scale.

# Materials and methods

**Key resources table**

| Reagent type (species) or resource | Designation | Source or reference | Identifiers | Additional information |
|---|---|---|---|---|
| Cell line (*Homo sapiens*) | HeLa | Jiadong Wang Laboratory | RRID:CVCL_0030 | |
| Cell line (*H. sapiens*) | SKOV3 | Jiadong Wang Laboratory | RRID:CVCL_0532 | |
| Cell line (*H. sapiens*) | MCF7 | Jiadong Wang Laboratory | RRID:CVCL_0031 | |
| Cell line (*H. sapiens*) | HEK293T | Jiadong Wang Laboratory | RRID:CVCL_0063 | |
| Cell line (*H. sapiens*) | SH-SY5Y | Jian Chen Laboratory | RRID:CVCL_0019 | |
| Cell line (*H. sapiens*) | VCaP | iCell Bioscience | RRID:CVCL_2235 | |
| Cell line (*H. sapiens*) | HEK293 | iCell Bioscience | RRID:CVCL_0045 | |
| Commercial assay or kit | Total RNA of human brain | Clontech | Cat. #: 636530 | |
| Commercial assay or kit | Total RNA of human testis | Clontech | Cat. #: 636533 | |

## Cell culture and RNA isolation

The human cell lines HeLa, SKOV3, MCF7, VCaP, SH-SY5Y, HEK293T, and HEK293 were used in this study. HeLa, SKOV3, MCF7, and HEK293T cell lines were obtained from Jiadong Wang Laboratory (*Xu et al., 2020*). SH-SY5Y cell line was obtained from Jian Chen Laboratory. VCaP and HEK293 cell

lines were purchased from iCell Bioscience. Cell lines were cultured in DMEM (Invitrogen) supplemented with 10% fetal bovine serum (YEASEN) and 1% penicillin/streptomycin (Solarbio) at 37°C with 5% $CO_2$. Cell lines were collected at 80–90% confluency. All cell lines were authenticated with STR profiling and tested negative for mycoplasma contamination. Total RNA was extracted by the FastPure Cell/Tissue Total RNA Isolation Kit (Vazyme) according to the manufacturer's instructions. Total RNA from the human brain (#636530) and testis (#636533) was purchased from Clontech.

## circRNA enrichment for library construction of RNA-seq and circFL-seq

### rRNA depletion

Similar to a previous method (*Morlan et al., 2012*), nonoverlapping synthetic DNA probes of complementary sequences (*Supplementary file 7*) of 18S and 28S rRNA at a final concentration of 1 μM as well as 5S and 5.8S rRNA, 12S and 16S mtrRNA, ETS, and ITS at a final concentration of 0.1 μM were pooled. About 1 μl of DNA probe was mixed with 2 μg of total RNA and 2 μl of hybridization buffer (750 mM Tris-HCl, 750 mM NaCl) at a final reaction volume of 15 μl. The mixture was heated to 95°C for 2 min, slowly cooled to 22°C (– 0.1°C/s), and incubated for an additional 5 min at 22°C. Also, 2 μl of thermostable RNase H (NEB) was added along with 2 μl of 10× RNase H buffer at a final reaction volume of 20 μl, incubated at 50°C for 30 min, and placed on ice. DNA probes were removed by 2.5 μl DNase I (NEB) with 3 μl 10× DNase I buffer at a final volume of 30 μl, incubated at 37°C for 30 min, and placed on ice.

### Poly(A) tailing

The reaction above was then mixed with 4 μl ATP (10 mM), 4 μl 10× Poly(A) Polymerase Reaction Buffer and 1 μl Poly(A) Polymerase (NEB) for poly(A) tailing and incubated at 37°C for 10 min. Purified RNA was isolated by 2.5× RNA Clean Beads (Vazyme) according to the manufacturer's instructions.

### RNase R treatment

For circFL-seq, RNA was incubated at 37°C for 30 min and 70°C for 10 min in a 10 μl reaction that contained 1 U RNase R (Lucigen) and 1 μl 10× RNase R buffer. For RNA-seq, RNA was incubated under the same conditions but in a 20 μl reaction containing 2 U RNase R and 2 μl 10× RNase R buffer.

## Full-length circRNA cDNA preparation for circFL-seq

### Reverse transcription

After enrichment of circRNAs, collected RNAs were reverse transcribed into first cDNA strands in a 20 μl reaction by P1-N6 (5'-GTCGACGGCGCGCCGGATCCATANNNNNN-3') with HiScript III reverse transcriptase (Vazyme) for 5 min at 25°C, 50 min at 50°C, 2 min at 70°C, and 5 s at 85°C, followed by purification with 0.75× DNA Clean Beads (Vazyme) according to the manufacturer's instructions.

### Poly(A) tailing

Then, poly(A) tails were added at the 3' ends in a 20 μl reaction by terminal deoxynucleotidyl transferase (Invitrogen) with final dATP and ddATP concentrations of 2.5 mM and 25 μM, respectively, followed by purification with 0.75× DNA Clean Beads.

### Second-strand synthesis

Next, second-strand cDNAs were synthesized using P2-T24 (5'-ATATCTCGAGGGCGCGCCGG ATCCTTTTTTTTTTTTTTTTTTTTTTTT-3') by I-5 High-Fidelity DNA polymerase (MCLAB) at 98°C for 2 min, 50°C for 2 min, and 72°C for 5 min.

### Amplification

Then, cDNAs were equally split into four 50 μl PCRs with primers P1 (5'-GTCGACGGCGCGCCGG ATCCATA-3') and P2 (5'-ATATCTCGAGGGCGCGCCGGATCC-3') and amplified by 20 cycles of 98°C for 10 s, 67°C for 15 s, and 72°C for 75 s, followed by 0.5× DNA Clean Bead purification. Approximately 10–50 ng purified cDNAs were further amplified by 8–10 cycles in a set of four 50 μl PCRs with P1 and P2, followed by purification with 0.5× DNA Clean Beads.

## Nanopore library construction and sequencing

A DNA library for barcoding and ligation sequencing was prepared following protocols EXP-NBD104 and SQK-LSK109. Briefly, 0.5–1 µg of circFL-seq cDNA was repaired and dA-tailed by NEBNext FFPE DNA Repair Mix (NEB) and NEBNext Ultra II End repair/dA-tailing Module (NEB), followed by purification with DNA Clean Beads. For multiplexing, repaired and end-prepped DNA was barcoded with Native Barcode by NEB Blunt/TA Ligase Master Mix (NEB), followed by purification with DNA Clean Beads. Barcoded samples were pooled together in equimolar amounts. Single samples without barcodes or pooled barcoded samples (700 ng) were ligated to an ONT Adapter with NEBNext Quick Ligation Module (NEB), followed by purification with DNA Clean Beads. The DNA library was mixed with sequencing buffer, and beads were loaded onto a PromethION or MinION R9.4 Flow Cell and run on a PromethION (performed by Grandomics) or MinION sequencer, respectively.

## cDNA library preparation and sequencing of RNA-seq

After circRNA enrichment, RNA was isolated by 2.5× RNA Clean Beads (Vazyme). cDNA libraries were constructed and barcoded by the VAHTS Universal V6 RNA-seq Library Prep Kit for Illumina (Vazyme) and VAHTS RNA Adapters set1/set2 for Illumina (Vazyme) according to the manufacturer's instructions for the strand-specific rRNA depletion library. The cDNA library was fragmented, and an insert size of approximately 300 bp was selected. Eight libraries were pooled together and sequenced on the Illumina HiSeq X Ten platform of Annoroad Gene Technology with a paired-end read length of 150 bp.

## Computational analysis of full-length circRNAs with circFL-seq

Raw sequenced data in fast5 format files were transformed to FASTQ format files (kept reads with qscore ≥7.0) and were demultiplexed by guppy (4.2.2). We employed porechop (v0.2.4) to trim barcode and circFL-seq primers (P1 and P2) and split chimeric reads to obtain clean reads for each sample. Then, RG mode (https://github.com/yangence/circfull, *Liu et al., 2021*) was employed to detect circRNAs with clean reads. cRG mode was used to correct circFL-seq reads by de novo self-correction to calculate CS, and RG mode was rerun with a query sequence of three copies of CS. With primer sequences at both ends of sequenced reads, circFL-seq detected the strand origin of reads and adjusted circRNA results with strand information. Finally, we filtered out low-quality full-length circRNAs after integrating circRNA results.

### RG mode

Clean reads were mapped to the human genome (hg19) by minimap2 (*Li, 2018*) (v2.12) with the parameters '-ax splice -p 0.5' or '-ax splice -p 0.5 -uf' for transcript-strand reads. Aligned reads with chiastic overlapping segments were recognized as candidate circRNA reads (CCRs). CCRs were classified into three types: normal, fusion on the same chromosome, and fusion on different chromosomes. The boundaries of the chiastic segment of the CCRs were detected as potential BSJs. FSJs were determined according to the skipped region from the reference. For each read, a pseudo reference sequence was generated by concatenating two sequences from 150 nt upstream to 150 nt downstream of the BSJ region. Then, CCRs were realigned against the pseudo references, and accurate sites of BSJs were determined by integrating multiple aligned BSJs, gene annotation (GENCODE v19), and canonical splicing motifs (GT/AG). FSJs of CCRs were also corrected based on the integration of multiple aligned FSJs, gene annotation, and FSJs from other CCRs with the same BSJ. Next, the full-length circRNA was constructed based on the FSJ(s) and BSJ(s). Incorrect construction of full-length circRNAs caused by mistaken alignment against tandem repeat sequences of the reference genome was identified by TideHunter (v1.0) with parameters '-f 2 –c 1.2 l', and these circRNAs were filtered out.

### The de novo self-correction

The CS of clean reads was detected by TideHunter (v1.0) with the parameters '-f 2 c 1.5 p 30 l'. Following evaluation by Tandem Repeats Finder (v4.09) with parameters '2 5 7 80 5 5 2000 h -ngs', CS was removed if containing internal tandem repeats, defined by an alignment score >40 or length of internal tandem repeats longer than half of the CS.

### cRG mode

To locate the genomic region of CS, pseudo query sequences from three combined monomers of a CS were created. The position of full-length circRNA was detected with RG mode.

### Identification of the strand origin of clean reads

CCRs with FSJs detected were first selected to determine the strand origin of sequenced reads, that is, first or second strand, based on the GT/AG motif of the FSJ and mapped strand. For each 100 nt flanking end of the raw read, the maximum identical number of bases to primers P1, P2, T24, and their reverse complementary sequences was calculated by the Smith-Waterman algorithm. With these numbers of identical bases as predictor variables and strand origin as the target variable, a random forest classifier was trained on 75% selected CCRs to predict the strand origin of all clean reads. The performance of the classifier was evaluated on the remaining 25% CCRs by accuracy and AUC value for each sample (*Supplementary file 8*). CircRNA with ambiguous strand direction was adjusted by rerunning RG mode with stranded reads.

### Filtration of low quality circRNA

After RG detection supplemented with cRG detection, low-quality circRNAs with an unsplicing ratio of BSJ <0.1 were filtered out. CircRNA isoforms with reliable BSJs and FSJs were retained if more than half of the circRNA reads were perfectly aligned on ±4 bp of junctions. CircRNA BSJs were filtered out if both BSJs were located in ±30 bp of repeat elements.

## Reverse transcription for PCR and RT-qPCR validation

For PCR validation, sample RNA was reverse transcribed to cDNA products for 5 min at 25°C, 50 min at 50°C, 2 min at 70°C, and 5 s at 85°C with random hexamers by HiScript III. For RT-qPCR, sample RNA was reverse transcribed to cDNA products for 15 min at 37°C and 5 s at 85°C with random hexamers and oligo(T) according to the manufacturer's instructions.

## Validation of full-length sequences of circRNA

A pair of divergent PCR primers was designed and included 6–8 additional G bases at the 5' end (*Supplementary file 9*) to reduce the accumulation of short PCR products. With cDNA products of the RT or circFL-seq library as template, PCR amplification in a 25 μl reaction of Takara Ex Taq Hot Start was carried out by the following program: 98°C for 10 s followed by 3 cycles of 98°C for 10 s, 60°C for 30 s, and 72°C for 90 s, then 30 cycles of 98°C for 10 s and 72°C for 90 s, with a final extension at 72°C for 60 s. The PCR products were analyzed on 1.2% agarose gels (TSINGKE), and the rolling circle bands were cut out and extracted, followed by TA cloning to a pEASY-T1 cloning vector (TransGen). Then, the clones with inserts were Sanger sequenced.

## Validation of AS in circRNAs

Total RNA of the HeLa cell line (1 μg) was treated with RNase R (4 U) in a 10 μl reaction and then reverse transcribed to the cDNA products. For each circRNA isoform from the same BSJ, divergent primers (*Supplementary file 10*) targeting but not across splicing junctions were used to validate the AS junction site in a PCR volume of 25 μl. The PCR products were analyzed on 1.2% agarose gels, and the target bands were cut out and extracted, followed by TA cloning and Sanger sequencing.

## Quantification of circRNA expression by RT-qPCR

Total RNA of HeLa and SKOV3 cell lines (1 μg) w/wo RNase R treatment (4 U) was reverse transcribed for RT-qPCR. For specific circRNAs, primers (*Supplementary file 11*) across a BSJ with cDNA as template in a 20 μl reaction were set according to the manufacturer's instructions for ChamQ Universal SYBR qPCR Master Mix (Vazyme). For specific circRNA isoforms, we designed primers (*Supplementary file 12*) for both specific BSJs and alternative FSJs in the reaction. For the quantification of internal AS events, two major circular isoforms 1 and 2 from the same BSJ were selected for the evaluation of the transcript ratio with the following formula: $\frac{2^{CT_{isoform1,2}}}{2^{CT_{isoform1}}+2^{CT_{isoform2}}}$ . Primers targeting *GAPDH* as a reference gene were used. Thermal cycling was carried out on an Applied Biosystems 7500 Fast system at 95°C for 5 min, followed by 40 cycles of 95°C for 10 s and 60°C for 24 s.

### Validation of f-circRNA

Total RNA of the MCF7 cell line (1 μg) w/wo RNase R treatment (4 U) was reverse transcribed for PCR. We validated full-length f-circRNA with primers (*Supplementary file 13*) for the two fusion junctions by RCRT and Sanger sequencing. Linear RNA was isolated to determine the origin of fusion junctions from BS or gene fusion. Because of the uncertainty of the poly(A) tail in linear RNA, total RNA was first treated with poly(A) tailing (NEB), and then linear RNA was selected by the poly(A) mRNA magnetic isolation module (NEB) according to the manufacturer's instructions. PCR was performed with the same primers of both fusion junctions of f-circRNA. For quantification of the fusion junction of f-circRNA, total RNA of the MCF7 cell line (1 μg) w/wo RNase R treatment (4 U) was reverse transcribed for RT-qPCR. Primers (*Supplementary file 14*) of target fusion junctions were designed for RT-qPCR.

### CircRNA analysis from RNA-seq

RNA-seq data were aligned to the human reference genome (hg19) by BWA (*Li and Durbin, 2009*) (v0.7.17-r1188). CircRNA BSJs were detected and quantified by CIRI2 with gene annotation (GENCODE v19). Full-length circRNA structures were constructed by CIRI-AS (*Gao et al., 2016*), CIRI-full (*Zheng et al., 2019*), and CIRI-vis (*Zheng and Zhao, 2020*). For fusion junctions, we directly searched the ±10 bp junction site in RNA-seq reads.

Analysis of isoCirc and CIRI-long data, isoCirc nanopore sequencing data were downloaded from the Sequence Read Archive (SRA: SRP235284). CircRNAs were analyzed with the isoCirc computational pipeline (https://github.com/Xinglab/isoCirc, *Xin et al., 2021b*) with the human reference genome (hg19) and gene annotation (GENCODE v19). CIRI-long nanopore sequencing data were downloaded from Genome Sequence Archive (GSA: CRA003317). CircRNAs were analyzed with the CIRI-long computational pipeline (https://github.com/bioinfo-biols/CIRI-long, *Zhang et al., 2021b*) with the mouse reference genome (mm10) and gene annotation (GENCODE vM23).

### Identification of differentially expressed circRNA

CircRNA BSJs with at least 10 read counts in two or more RNA-seq samples and two or more circ-FL-seq samples of either the HeLa or SKOV3 cell line were kept to identify differentially expressed BSJs. DESeq2 with 'mean' fitType was employed to analyze differential expression between HeLa and SKOV3 cells. Differential BSJs at an FDR<0.05 were recognized as DECs.

### Data availability

The circFL-seq and RNA-seq data produced by this study have been deposited in SRA (PRJNA722575). Information on circRNAs detected by circFL-seq is available in the figshare repository (https://doi.org/10.6084/m9.figshare.14265650.v1). The computational software of circFL-seq can be accessed from https://github.com/yangence/circfull, (*Liu et al., 2021*) copy archived at swh:1:rev:2e8be5c116e9c226661c63e35223f2120272ccfc.

## Acknowledgements

The work was supported by grants from the Beijing Municipal Science and Technology Commission of China (7212065), the Chinese Institute for Brain Research, Beijing (2020-NKX-XM-01), and the Beijing Municipal Science and Technology Commission of China (Z181100001518005).

## Additional information

### Funding

| Funder | Grant reference number | Author |
|--------|------------------------|--------|
| Beijing Municipal Science and Technology Commission | 7212065 | Ence Yang |
| Chinese Academy of Sciences | 2020-NKX-XM-01 | Ence Yang |

| Funder | Grant reference number | Author |
|---|---|---|
| Beijing Municipal Science and Technology Commission | Z181100001518005 | Ence Yang |

The funders had no role in study design, data collection and interpretation, or the decision to submit the work for publication.

## Author contributions

Zelin Liu, Conceptualization, Data curation, Formal analysis, Investigation, Methodology, Resources, Software, Validation, Visualization, Writing - original draft; Changyu Tao, Data curation, Investigation; Shiwei Li, Yongtai Bai, Resources, Validation; Minghao Du, Data curation, Resources; Xueyan Hu, Resources, Writing - review and editing; Yu Li, Resources; Jian Chen, Resources, Supervision; Ence Yang, Conceptualization, Funding acquisition, Project administration, Supervision, Writing - original draft

## Author ORCIDs

Zelin Liu (iD) http://orcid.org/0000-0002-3516-3999
Ence Yang (iD) http://orcid.org/0000-0002-9526-2737

## Decision letter and Author response

Decision letter https://doi.org/10.7554/eLife.69457.sa1
Author response https://doi.org/10.7554/eLife.69457.sa2

# Additional files

## Supplementary files

- Supplementary file 1. Data summary of circFL-seq library.
- Supplementary file 2. Summary of alternative splicing events of circRNAs detected by circFL-seq.
- Supplementary file 3. Data summary of RNA-seq library.
- Supplementary file 4. Comparison of isoCirc and circFL-seq for circRNA detection in the HEK293 cell line.
- Supplementary file 5. Computational analysis of circFL-seq and CIRI-long.
- Supplementary file 6. Comparisons between circFL-seq, CIRI-long, and isoCirc.
- Supplementary file 7. Sequences of hybrid probes for rRNA degradation.
- Supplementary file 8. Summary of performance of strand classifier.
- Supplementary file 9. Primers to validate rolling circles of circRNAs.
- Supplementary file 10. Primers to validate alternative splicing of circRNAs.
- Supplementary file 11. Primers to validate the expression levels of circRNA BSJs by RT-qPCR.
- Supplementary file 12. Primers to validate the expression levels of circRNA isoforms by RT-qPCR.
- Supplementary file 13. Primers to validate full-length sequence of f-circRNA.
- Supplementary file 14. Primers to validate the expression levels of f-circRNA junctions by RT-qPCR.
- Transparent reporting form

## Data availability

The circFL-seq and RNA-seq data produced by this study have been deposited in SRA (PRJNA722575). The information of circRNAs detected by circFL-seq is available in the figshare repository (https://doi.org/10.6084/m9.figshare.14265650.v1). The computational software circfull can be accessed from https://github.com/yangence/circfull (copy archived at https://archive.softwareheritage.org/swh:1:rev:2e8be5c116e9c226661c63e35223f2120272ccfc).

The following dataset was generated:

| Author(s) | Year | Dataset title | Dataset URL | Database and Identifier |
|---|---|---|---|---|
| Liu ZL | 2021 | circRNA_circFL_table.xlsx | https://doi.org/10.6084/m9.figshare.14265650.v1 | figshare, 10.6084/m9.figshare.14265650.v1 |
| Liu ZL | 2021 | circFL-seq, a full-length circRNA sequencing method | https://www.ncbi.nlm.nih.gov/bioproject/PRJNA722575 | NCBI BioProject, PRJNA722575 |

The following previously published datasets were used:

| Author(s) | Year | Dataset title | Dataset URL | Database and Identifier |
|---|---|---|---|---|
| Xin R, Gao Y, Gao Y, Wang R, Kadash-Edmondson KE, Xing Y | 2021 | isoCirc catalogs full-length circular RNA isoforms in human transcriptomes | https://www.ncbi.nlm.nih.gov/geo/query/acc.cgi?acc=GSE141693 | NCBI Gene Expression Omnibus, GSE141693 |
| Zhang J, Hou L, Zuo Z, Ji P, Zhang X, Xue Y, Zhao F | 2021 | Comprehensive profiling of circular RNAs with nanopore sequencing and CIRI-long | https://ngdc.cncb.ac.cn/gsa/browse/CRA003317 | NGDC Genome Sequence Archive, CRA003317 |

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
