## [Decision Letter]

**Acceptance summary:**

Yang and colleagues report circFL-seq a method for sequencing full length circular RNAs with rolling circle RT and nanopore sequencing. While two other methods have recently been published, this manuscript does add to a growing literature on long read sequencing of circular RNAs.

**Decision letter after peer review:**

Thank you for submitting your article "circFL-seq reveals full-length circular RNAs with rolling circular reverse transcription and nanopore sequencing" for consideration by *eLife*. Your article has been reviewed by 3 peer reviewers, and the evaluation has been overseen by a Reviewing Editor and James Manley as the Senior Editor. The following individual involved in review of your submission has agreed to reveal their identity: Fangqing Zhao (Reviewer #3).

Essential revisions:

1. Reviewers all agree that a major weakness of the present manuscript is with the comparison to existing methods. The authors should compare circFL-seq to CIRI-long, and reviewers agree that a three way comparison to isoCirc is informative. The authors should accompany their comparisons with a discussion about strengths and weaknesses of each of the methods.

2. The current claims and conclusions that circFL-seq is the superior method is not well supported by their data in this version of the manuscript. The authors should provide fair critiques of the other method (see comments about ligation). The authors need to point out potential limitations. For example, Is CircFL-seq biased to detecting only highly expressed transcripts?

3. In comparison to isoCirc, circFL-seq identified fewer circRNA isoforms with higher read coverage of the detected circRNAs, which may be a PCR artifact. The authors need to address this concern. In addition, in two published studies, isoCirc and CIRI-long have also used nanopore sequencing to characterize circRNA isoforms and alternative splicing events. However, both studies have reported a relatively higher percentage of retained introns (isoCirc: Figure 4b, CIRI-long: Supplementary Figure 13) compared to the number of 3.5% of intron retention events in line 139. The authors should clarify the reason behind this difference.

4. The authors found six f-circ derived from GBF1-MACROD2 fusion and validated their junctions using Sanger sequencing. Besides, the authors also used short-read RNA-seq to validate the linear fusion junctions. What's the ratio of linear and circular transcript derived from these gene fusion loci? Is there any possibility that these f-circRNAs are derived from trans-splicing events? Considering that short-read RNA-seq data cannot effectively distinguish circular and linear transcripts, the authors may try to search for nanopore reads spanning the fusion region, which can provide direct evidence for these gene fusion events.

5. Because of the high error rate of nanopore sequencing, the authors should compare the error rate of CS sequence before and after cRG correction to elucidate the ability to correct sequencing errors with the cRG mode.

6. The authors trained a random forest classifier to predict the strand origin of circular reads. How many CCRs were used as the training set, and how's the performance of the random forest classifier? The authors should provide more data about this step.

General comments:

1. The Figures and Supplemental Figures were not labeled with figure numbers, making it extremely difficult to read (especially some figures span across more than one page). The authors should label the figure number more clearly.

2. Some figures are not clearly labeled. For example, in Figure 2j, what does each lane represent? Also, in Figure 3e, it is not clear what the y-axis is. The authors state that it is read coverage, but how come the value is from 0 to 1? In Figure 3d, the authors state that it is the correlation between HeLa and SKOV3, but it was not clear what axis is HeLa and SKOV3, respectively. Again, what does the shaded area mean in these figures (figure 3c, 3d, 3f, and 3g)? The authors should check through the figure label more carefully and correct them accordingly.

*Reviewer #1 (Recommendations for the authors):*

Overall, this is a nice piece of work that can be published after revision. Below are my specific comments for the authors:

1. As noted in my public review, the authors should carry out a two-way comparison between circFL-seq and CIRI-long, as well as a three-way comparison between circFL-seq, CIRI-long, and isoCirc. Given that circFL-seq and CIRI-long are both based on RCRT while isoCirc is based on RCA, it would be interesting to see if circFL-seq and CIRI-long produce more concordant results in terms of the discovery and quantitation of full-length circular RNAs, given the similarity in their experimental strategies.

2. In both Introduction and Discussion, the authors noted that the use of ligation in isoCirc may lead to false discoveries of circular RNAs. While this statement is technically correct from an experimental standpoint, such false discoveries can be recognized and removed computationally – therefore this is not a fair critique of isoCirc. In fact, the isoCirc computational pipeline is designed to remove such artifacts, using stringent requirements for alignment quality and presence of canonical splice site motifs in all forward and back splice junctions within full-length circular RNA transcripts.

3. Line 65-67: "Thus, an accurate but affordable method to detect full-length circRNA remains to be developed for wide application in screening functional circRNAs at the omics scale". This statement leaves the impression that such a method currently does not exist, which is not a fair representation of the current literature with the recent publications of isoCirc and CIRI-long.

4. The authors reported that compared to isoCirc, circFL-seq produced more full-length circular RNA reads at the same library depth but identified fewer circular RNA isoforms. The authors appeared to present this finding as a positive feature of circFL-seq. For example, in line 265-267, the authors stated that "as a trade-off, isoCirc produced fewer reads with the same sequencing depth, which raised sequencing costs and weakened its ability to detect and accurately quantify high-quality circRNAs". There are multiple issues with this statement. In terms of the precision of circular RNA quantitation (e.g. as evaluated based on comparison among nanopore replicates as well as comparison to short-read RNA-seq data), the metrics presented for circFL-seq are in fact quite comparable to the metrics presented in the isoCirc paper, so there is no evidence that circFL-seq provides a better quantitation of circular RNAs. Moreover, given that the ground-truth is not known, an alternative interpretation to this observation is that circFL-seq may be biased towards highly expressed circular RNAs, and may lack the ability to discover moderately and lowly expressed circular RNAs. Overall, in comparing different methods, the authors should aim to provide an impartial discussion about the strengths and weaknesses of individual methods, and avoid over-interpreting the data in favor of their own method.

5. The discussion about CDR1as is interesting. Can CIRI-long detect this circular RNA?

6. The authors should also cite the BioRxiv preprint by Rahimi et al., (https://doi.org/10.1101/567164), which is another method for nanopore sequencing of circular RNAs.

7. Overall, the manuscript is easy to read and follow, but it could benefit from a thorough editing by a language editor.

*Reviewer #2 (Recommendations for the authors):*

1. In comparison to isoCirc, circFL-seq identified fewer circRNA isoforms with higher read coverage of the detected circRNAs. This raises a concern that the outcome may result from that RCRT captures circRNA less efficiently than RCA, resulting in fewer circRNA molecules are captured in circFL-seq. The higher read coverage may simply come from sequencing the same circRNA molecule from the PCR amplification artifacts. This may also explain why circFL-seq cannot detect circRNAs with low read count or lowly expressed circRNAs. In this case, the authors cannot use back splice junction (BSJ) detection saturation as an indicator to compare the required read-depth between isoCirc and circFL-seq. Also, given the concern above, the "high read coverage" does not necessarily mean "high quality" nor "high accuracy" as claimed by the authors. The authors should address this concern before claiming on the benefits of high read coverage.

2. The advantages of circFL-seq over other existing technologies are not well-supported. For example, the authors claim that RCRT has lower residual linear RNA contamination than RCA, but the authors do not provide any data or evidence supporting the claim. Also, the authors claim that the circFL-seq gives higher circRNA read coverage; hence it is beneficial for circRNA quantification. However, the real "benefits" over other technologies (RNA-seq and isoCirc) for circRNA quantification are not clear since the RNA-seq (and isoCirc) quantification is significantly correlated with circFL-seq as demonstrated by the authors.

3. In the manuscript, the results are often comparable with known database and existing technologies when the authors focus on the "high quality" circRNAs only (circFL-seq read counts >= 5) which also have high expression level. The fact suggests that circFL-seq result is trust-worthy on "high quality" only. It also suggests that circFL-seq may fail to detect the lowly expressed circRNAs. The authors should note and discuss these limitations.

General comments:

1. The Figures and Supplemental Figures were not labeled with figure numbers, making it extremely difficult to read (especially some figures span across more than one page). The authors should label the figure number more clearly.

2. Some figures are not clearly labeled. For example, in Figure 2j, what does each lane represent? Also, in Figure 3e, it is not clear what the y-axis is. The authors state that it is read coverage, but how come the value is from 0 to 1? In Figure 3d, the authors state that it is the correlation between HeLa and SKOV3, but it was not clear what axis is HeLa and SKOV3, respectively. Again, what does the shaded area mean in these figures (figure 3c, 3d, 3f, and 3g)? The authors should check through the figure label more carefully and correct them accordingly.

Specific comments:

1. In the Abstract section, the authors claim that "… the detection of cancer-related fusion circRNAs…". However, the authors did not provide any data or literature suggesting that the fusion circRNAs they identified by circFL-seq are really "cancer-related" or biologically meaningful. The claim needs to be revised or proved.

2. In Figure2—figure supplement 1a, the full-length circRNA reads contribute very little percentage of the total clean reads (~2-5%). How come the RCRT method generate so little full-length circRNA reads? The authors should comment on this and discuss it.

3.In Figure 3—figure supplement 3, the author claim that more reads are required for ONT to confidently identify circRNAs. Doesn't this compromise the cost-efficient claim of circFL-seq made by the authors earlier? The authors should comment on this and discuss it.

4. In Figure 3—figure supplement 6a-d, how the saturation curves are calculated? It seems like isoCirc has much lower sequencing depth (~0.3 M) than circFL-seq. How do the authors compare the BSJ saturation in this case?

5. When comparing the validity of circFL-seq and isoCirc, why do the authors focus on top 100 expressed BSJ only? A better comparison should be the total BSJ in circFL-seq and isoCirc.

6. The authors should not use the same absolute circFL-seq BSJ read counts to define high-quality BSJs in isoCirc for the following reasons: (i) the read counts >= 5 in circFL-seq is arbitrary, there is no evidence suggesting that the isoCirc should use the same read counts to define high-quality BSJs. (ii) Since isoCirc captures more circRNAs, a lower BSJ read counts per circRNA is expected given the same sequencing depth. In both cases, lower BSJ read counts in isoCirc does not necessarily mean the BSJ is not "high-quality". Thus, the authors should not use absolute circFL-seq BSJ read counts as an indicator for the BSJ quality in isoCirc.

7. In the PLOD2 circFL-seq and RNA-seq example shown by the authors, the authors suggest that the circPLOD2 has lower exon skipping event than its parent linear RNA in HeLa cells. However, given that the BSJ is exactly the same between exon-skipped and non-exon-skipped circPLOD2, how does the back-splicing mechanism distinguish different parent linear RNA isoforms that selectively back-splices the non-exon-skipped linear RNA to generate specific circPLOD2 isoform in HeLa cells?

8. Are the f-circ detected by circFL-seq generated by RNA fusion or genomic fusion? Although the RNA-seq suggests a genomic fusion, it does not completely eliminate the possibility of a genomic fusion. A genomic PCR followed by Sanger sequencing should be performed to validate the fusion junction of the genome.

*Reviewer #3 (Recommendations for the authors):*

Considering that there are two recently published circRNA reconstruction tools based on nanopore sequencing, the authors should comprehensively compare their method with these two tools, and carefully discuss the advantages and disadvantages of these methods.

Specific comments:

1. In the Discussion section (line 260-262), the authors compared circFL-seq with the recently published CIRI-long method. Both circFL-seq and CIRI-long use a similar rolling circle reverse transcription strategy to amplify circRNAs. The authors may discuss the difference and (dis)advantages between their method and previous methods (isoCirc and CIRI-long).

2. In section "Comparison with RNA-seq and isoCirc for circRNA detection", the authors compared circFL-seq with the isoCirc method and found that circFL-seq produced more circular reads but identified fewer circRNA isoforms, which is an interesting result. Does it mean that isoCirc has a better sensitivity or higher false discovery rate in detecting lowly expressed circRNAs? The authors should include more comparison (e.g. venn diagram between three sets under different BSJ thresholds) between circFL-seq, isoCirc, and public circRNA database (e.g. circAtlas [PMID: 32345360, PMID: 30893614]) to demonstrate the advantages of their method.

3. The authors found six f-circ derived from GBF1-MACROD2 fusion and validated their junctions using Sanger sequencing. Besides, the authors also used short-read RNA-seq to validate the linear fusion junctions. What's the ratio of linear and circular transcript derived from these gene fusion loci? Is there any possibility that these f-circRNAs are derived from trans-splicing events? Considering that short-read RNA-seq data cannot effectively distinguish circular and linear transcripts, the authors may try to search for nanopore reads spanning the fusion region, which can provide direct evidence for these gene fusion events.

4. Because of the high error rate of nanopore sequencing, the authors should compare the error rate of CS sequence before and after cRG correction to elucidate the ability to correct sequencing errors with the cRG mode.

5. The authors trained a random forest classifier to predict the strand origin of circular reads. How many CCRs were used as the training set, and how's the performance of the random forest classifier? The authors should provide more data about this step.

6. In section "Evaluation of quantification of full-length circRNAs", it would be nice if the authors could compare the quantification results between their method on nanopore reads and previous method (e.g. CIRIquant) on short-read RNA-seq data.

7. The authors used different names (circFL-seq, circfull) to denote their sequencing and data analysis methods. It would be better if they can unify the name, say, circFL-seq, which may avoid misunderstanding.

---

## [Author Response]

Essential revisions:1. Reviewers all agree that a major weakness of the present manuscript is with the comparison to existing methods. The authors should compare circFL-seq to CIRI-long, and reviewers agree that a three way comparison to isoCirc is informative. The authors should accompany their comparisons with a discussion about strengths and weaknesses of each of the methods.

We agree that comparison among the three methods will facilitate selecting proper method for different purposes. However, detection of full-length circRNAs is affected by various confounding factors (as summarized in Author response table 1), including input of total RNA, species of samples, libraries per sample, sequencing platform and libraries per Flow Cell. As these factors were not consistent among the three methods developed at almost the same time, we compared these methods in principle.

**Author response table 1. sa2table1:** Summary of confounding factors among three methods.

details	circFL-seq	CIRI-long	isoCirc
input of total RNA (μg)	2	1	20
species	Human	mouse	human
samples	7 cell linesbrain and testis	brain	1 cell line12 tissues
platform	ONT PromethION, MinION	ONT MinION	ONT MinION
libraries per sample	one/two	multiple	multiple
libraries per Flow Cell (sequencing depth)	one/multiple	multiple	one

All the three methods produced rolling circles of circRNA for full-length sequencing by ONT. However, circFL-seq and CIRI-long employed RCRT strategy, while isoCirc with RCA strategy. Benefiting from RCA with circular cDNA template from ligation of reverse transcription (RT) product, isoCirc produced longer reads (up to 50 kb) containing more rolling circles than RCRT based methods (ca. 1 kb), and thus had an advantage in error correction during consensus sequences identification. As a trade-off, the cost of each full-length read is much higher in isoCirc than circFL-seq and CIRI-long. In addition, the unique ligation of cDNA in isoCirc may introduce false positive for circRNA detection by ligating cDNA from residual linear RNA. In contrast to isoCirc, both of circFL-seq and CIRI-long based methods employed RCRT to produce rolling circles during first-strand synthesis. The difference was that circFL-seq synthesized second-strand cDNA with an anchor primer while CIRI-long used template switching. Thus, these two methods are more resistant to the interference of residual linear RNA and generate shorter full-length reads of circRNA than isoCirc. Although decreasing sequencing cost of nanopore, shorter full-length reads have to perform error correction among reads besides intra-reads. Thus, RCRT based methods are not as sensitive as isoCirc with the same amount of full-length reads.

Along with detection of circRNA full-length sequences, all the three methods were also able to detect alternative splicing events of circRNAs. Both circFL-seq and isoCirc detected AS events of circRNA in human samples. For HEK293 cell line sequenced by one MinION, circFL-seq could be more precise than isoCirc for BSJ detection (75.3% vs 43.8%) with known circRNAs annotated in databases for comparison, and identified more than two times of ES, A3SS, and A5SS but similar amount of IR events than isoCirc. (Supplementary file 4). Thus, benefiting from more full-length circRNA reads, circFL-seq is more sensitive for AS events. However, circFL-seq may fail to detect some IR events. We found that the reads of isoCirc-only IR showed high error rate in the intron region (Author response image 1) which could due to the strong second structure. The RT step and cDNA ligation of isoCirc could avoid or remedy the early stop of RT in the strong RNA structure of intron.

**Author response image 1. sa2fig1:** Alignment schematic of nanopore reads of isoCirc-only IR events.

Although circFL-seq and CIRI-long were applied in different species (human vs. mouse), the computational pipeline was compatible and could be compared with each other for the very similar experimental protocols. We employed the computational pipeline of circFL-seq and CIRI-long to analyze HEK293 cell line of circFL-seq and mouse brain data of CIRI-long, respectively. We found the computational pipeline of circFL-seq is much more sensitive with similar precision for BSJ detection. For AS events, circFL-seq is also more sensitivity, especially for ES, A3SS and A5SS events. (Supplementary file 5).

We added above results and discussions in our revised manuscript and Supplementary file 4-6.

2. The current claims and conclusions that circFL-seq is the superior method is not well supported by their data in this version of the manuscript. The authors should provide fair critiques of the other method (see comments about ligation). The authors need to point out potential limitations. For example, Is CircFL-seq biased to detecting only highly expressed transcripts?

In the comparison with isoCirc on one MinION data of HEK293 cell line, circFL-seq is more precise with all read counts (both highly and lowly expressed transcripts). Thus, circFL-seq was not biased to only highly expressed transcripts (Author response table 2). The shortness of circFL-seq compared to isoCirc is that the RT step could early stop resulting in lack of IR detection from strong second structure of intron (see Author response image 1).

**Author response table 2. sa2table2:** Comparison of isoCirc and circFL-seq for BSJ detection in HEK293 cell line.

	**total circRNA BSJs**					
**# read counts for BSJ**	**1**	**2**	**3**	**4**	**>4**	**all**
isoCirc HEK293 SRR10612050	32,204	3,572	1,237	620	1,777	39,410
isoCirc HEK293 SRR10612051	34,493	3,916	1,361	687	1,915	42,372
isoCirc HEK293 SRR10612052	44,586	5,270	1,707	860	2,635	55,058
isoCirc HEK293 SRR10612053	39,484	5,274	1,871	1,022	2,970	50,621
isoCirc HEK293 SRR10612054	40,928	5,259	1,897	1,071	3,009	52,164
isoCirc HEK293 SRR10612055	30,647	3,779	1,387	710	2,024	38,547
isoCirc HEK293 all	158,875	23,302	8,821	5,133	26,782	222,913
circFL-seq HEK293	13,906	4,889	2,830	1,525	4,719	27,869
	**known circRNA BSJs annotated in database**					
**# read counts for BSJ**	**1**	**2**	**3**	**4**	**>4**	**all**
isoCirc HEK293 SRR10612050	10,458	2,528	1,080	571	1,620	16,257
isoCirc HEK293 SRR10612051	10,889	2,663	1,194	615	1,751	17,112
isoCirc HEK293 SRR10612052	12,727	3,447	1,451	773	2,396	20,794
isoCirc HEK293 SRR10612053	14,828	3,893	1,665	944	2,711	24,041
isoCirc HEK293 SRR10612054	15,078	3,834	1,678	971	2,750	24,311
isoCirc HEK293 SRR10612055	12,534	2,969	1,264	660	1,860	19,287
isoCirc HEK293 all	28,917	12,088	6,821	4,411	25,301	77,538
circFL-seq HEK293	8,836	3,821	2,377	1,365	4,589	20,988
	**% known circRNA BSJs**					
**# read counts for BSJ**	**1**	**2**	**3**	**4**	**>4**	**all**
isoCirc HEK293 SRR10612050	32.5	70.8	87.3	92.1	91.2	41.3
isoCirc HEK293 SRR10612051	31.6	68.0	87.7	89.5	91.4	40.4
isoCirc HEK293 SRR10612052	28.5	65.4	85.0	89.9	90.9	37.8
isoCirc HEK293 SRR10612053	37.6	73.8	89.0	92.4	91.3	47.5
isoCirc HEK293 SRR10612054	36.8	72.9	88.5	90.7	91.4	46.6
isoCirc HEK293 SRR10612055	40.9	78.6	91.1	93.0	91.9	50.0
isoCirc HEK293 all	18.2	51.9	77.3	85.9	94.5	34.8
circFL-seq HEK293	63.5	78.2	84.0	89.5	97.2	75.3

3. In comparison to isoCirc, circFL-seq identified fewer circRNA isoforms with higher read coverage of the detected circRNAs, which may be a PCR artifact. The authors need to address this concern. In addition, in two published studies, isoCirc and CIRI-long have also used nanopore sequencing to characterize circRNA isoforms and alternative splicing events. However, both studies have reported a relatively higher percentage of retained introns (isoCirc: Figure 4b, CIRI-long: Supplementary Figure 13) compared to the number of 3.5% of intron retention events in line 139. The authors should clarify the reason behind this difference.

Sorry for the misunderstanding. Compared to isoCirc, circFL-seq detected fewer circRNA BSJs. However, less than half of isoCirc detected BSJs were not annotated in database, while 75.3% of circFL-seq detected BSJs were annotated in databse (Author response table 2). Thus, circFL-seq produced more full-length reads for true positive circRNAs. We rephrased our manuscript.

Thank you for the inspiring question. We conducted a comprehensive investigation about AS detection among the three methods. Actually, the total amounts of IR events are comparable among the three methods. However, circFL-seq identified more ES, A3SS, and A5SS, resulting in the lower IR proportion. We added these data in the manuscript and Supplementary file 4,5.

4. The authors found six f-circ derived from GBF1-MACROD2 fusion and validated their junctions using Sanger sequencing. Besides, the authors also used short-read RNA-seq to validate the linear fusion junctions. What's the ratio of linear and circular transcript derived from these gene fusion loci? Is there any possibility that these f-circRNAs are derived from trans-splicing events? Considering that short-read RNA-seq data cannot effectively distinguish circular and linear transcripts, the authors may try to search for nanopore reads spanning the fusion region, which can provide direct evidence for these gene fusion events.

Based on the nanopore reads counts of f-circ and RT-qPCR results, we inferred that the ratio of circular and linear transcript of gene fusion loci was 9.9%. To verify these f-circ were circular transcripts rather than linear transcripts, we detected these f-circ in RNase R treated samples and validated by Sanger sequencing (Figure 4C-E in manuscript). In addition, we also found nanopore reads that contained more than 4 full circles of f-circ. Together, these evidences supported the *bona fide* fusion circRNAs from *GBF1-MACROD2*.

We thank the reviewers for the interesting question that the f-circ were derived from genomic fusion of *GBF1* and *MACROD2* or trans-splicing events between *GBF1* and *MACROD2*. Given that there were at least five f-circ isoforms of *GBF1-MACROD2*, we speculated that these f-circ were derived from genomic fusion rather than trans-splicing. However, length of the adjacent introns (83,935 bp for *GBF1* and 544,981 bp for *MACROD2*) of the fusion site were too large to detect genomic fusion by PCR. Long-read sequencing of genomic DNA could be helpful to identify the fusion junction and confirm the origin in the future.

A nanopore read with four circles of f-circ from *GBF1-MACROD2* E1-E2-E4

TCAAACGAAATGCCCGATGGAGCACCACTTTCCACTGGATGAAGAACGGGATCACAAACGCATAGTTTCGGTCATCTAAAGGAGGTTTTAAACAGTATAACAGTGGATGGCTGTATTCATAGAGCAGCCGGCCCCTGTTTGCTAGCTGAATGTCGTAACCTGAATGAAAACAGTGATACTGGACATGCAAAAATCACATGTGGCTATGACCTTCCTGCGAAATGTTTGCCAAGATGGTGGATAAGAATATTTACATCATTCAGGGGAGATTAACATTGTGGTTGGGGCCATCAAACGAAATGCCCGATGGAGCACCCATACACCACTGGATGAGAACGCCCCGGGATCCTCTGCTGCATAGTTTCGGTCATCTAAAGGAGGTTTTAAACAGTATAACAGTGGATGGCTGTATTCATAGAGCAGCCGGCCCCTGTTTGCTAGCTGAATGTCGTAACCTGAATGGCTGTGATACTGGACATGCAAAAATCACATGTGGCTATGACCTTCCTGCAAAATGTTTGCCAAGATGGTGGATAAGAATATTTACACATCATTCAAGGGGAGATTGCTGTGTGGTTGGGGCCATCAAACGAAATGCCCGATGGAGCACCCGCACACACTGGATGAAGAACGGGATCCTGCTGCATAGTTTCGGTCATCTAAAGGAGGTTTAAACAGTATAACAGTGGATGGCTGTATTCATAGAGCAGCCGGCCCCTGTTTGCTAGCTGAATGTCGTAACCTGAATGGCTGTGATACTGGACATGCAAAAATCACATGTGGCTATGACCTTCCTGCAAAATGTTTGCCAAGATGGTGGATAAGAATATTTACATCATTCAAGGAGATTAACATTGTGGTTGGGGCCATCAAACGAAATGCTTGACGGAACACCCATACACCACTGGATGAAGAACGCAGGATCCTCTGCTGCATAGTTTCGGTCATCTAAAGGAGGTTTTAAACAGTATAACAGTGGATGGCTGTATTCATAGAGCAGCCGGCCCCCTGTTGCTAGCTGAATGTCGTAACCTGAATGGCTGTGATACTGGACATGCAAAAATCACATGTGGCTATGACCTTCCTGCAAAATGTTTGCCAAGATTGGTGGATAAGAATATTTACATCATTCCGGGAGATTAACATTGTGGTTGGGGCCATCAAACGAAATGCCCGATGGAGCACCCGCGCCTACTGGATGAGCAAGGATCCTCTGCTGCATAGTTTCGGTCATCTAAAGGAGGTTTTAAACAGTATAACAGTGGATGGCTGTATTCATAGAGCAGCCGGCCCCTGTTTGCTAGCTGAATGTCGTAACCTGAATGGCTGTGATACTGGACATGCAAAAATCACATGTGTATGACCTTCCTGCAAAATGTTTCTGCCAGATGGTATGGATC

5. Because of the high error rate of nanopore sequencing, the authors should compare the error rate of CS sequence before and after cRG correction to elucidate the ability to correct sequencing errors with the cRG mode.

Thanks for the advice. With the suggested analysis, we found that cRG correction significantly reduced the error rate of both mismatch and indels. We have supplemented the information in our manuscript and Figure 2—figure supplement 2C, D.

6. The authors trained a random forest classifier to predict the strand origin of circular reads. How many CCRs were used as the training set, and how's the performance of the random forest classifier? The authors should provide more data about this step.

Thanks for the suggestion. We have supplemented details of the random forest classifier including the sample size for training and performance for predication (accuracy, AUC value). We have supplemented the information in our manuscript and Supplementary file 8.

General comments:1. The Figures and Supplemental Figures were not labeled with figure numbers, making it extremely difficult to read (especially some figures span across more than one page). The authors should label the figure number more clearly.

Sorry for the inconvenience. We have revised and labeled all figures and supplemental figures.

2. Some figures are not clearly labeled. For example, in Figure 2j, what does each lane represent? Also, in Figure 3e, it is not clear what the y-axis is. The authors state that it is read coverage, but how come the value is from 0 to 1? In Figure 3d, the authors state that it is the correlation between HeLa and SKOV3, but it was not clear what axis is HeLa and SKOV3, respectively. Again, what does the shaded area mean in these figures (figure 3c, 3d, 3f, and 3g)? The authors should check through the figure label more carefully and correct them accordingly.

Thank you for your careful review. We have labeled the lane name in Figure 2j. We have revised ‘read coverage’ to ‘adjusted coverage’ in Figure 3e, which was a ratio value normalized by max read coverage. In Figure 3d, the x-axis is the fold change of expression level between HeLa and SKOV3, the y-axis is the -∆∆CT of RT-qPCR between HeLa and SKOV3. The shaded areas in Figure 3c, 3d, 3f, 3g were 95% confidence intervals. Besides revisions in figures, we have added more details in figure legend (marked as G2).

Reviewer #1 (Recommendations for the authors):Overall, this is a nice piece of work that can be published after revision. Below are my specific comments for the authors:1. As noted in my public review, the authors should carry out a two-way comparison between circFL-seq and CIRI-long, as well as a three-way comparison between circFL-seq, CIRI-long, and isoCirc. Given that circFL-seq and CIRI-long are both based on RCRT while isoCirc is based on RCA, it would be interesting to see if circFL-seq and CIRI-long produce more concordant results in terms of the discovery and quantitation of full-length circular RNAs, given the similarity in their experimental strategies.

Thank you for your suggestion. We have performed the three-way comparison.

2. In both Introduction and Discussion, the authors noted that the use of ligation in isoCirc may lead to false discoveries of circular RNAs. While this statement is technically correct from an experimental standpoint, such false discoveries can be recognized and removed computationally – therefore this is not a fair critique of isoCirc. In fact, the isoCirc computational pipeline is designed to remove such artifacts, using stringent requirements for alignment quality and presence of canonical splice site motifs in all forward and back splice junctions within full-length circular RNA transcripts.

We agreed that false discoveries can be recognized and removed computationally. However, not all of those false discoveries were identified by the computational pipeline of isoCirc. We focused on the false discoveries of top expressed circRNAs to eliminate the effect of high noise of nanopore reads and disturbance of low expressed circRNAs. As clarified in our manuscript, when focusing on the top 100 expressed circRNA BSJs identified by each method, BSJs from both circFL-seq and RNA-seq (collected from isoCirc study) were annotated in databases, while 22 of the top 100 BSJs identified by isoCirc lacked evidence support from the database, RNA-seq, or circFL-seq results. Specifically, two products were from histone genes whose linear RNAs are resistant to RNase R degradation. In addition, we also failed to experimentally validate the two potential circular products with divergent primers. Thus, there are abundant false discoveries which were not removed by the computational pipeline of isoCirc. We have made this description clearer in our manuscript (marked as Q1.2).

3. Line 65-67: "Thus, an accurate but affordable method to detect full-length circRNA remains to be developed for wide application in screening functional circRNAs at the omics scale". This statement leaves the impression that such a method currently does not exist, which is not a fair representation of the current literature with the recent publications of isoCirc and CIRI-long.

We have revised this statement.

4. The authors reported that compared to isoCirc, circFL-seq produced more full-length circular RNA reads at the same library depth but identified fewer circular RNA isoforms. The authors appeared to present this finding as a positive feature of circFL-seq. For example, in line 265-267, the authors stated that "as a trade-off, isoCirc produced fewer reads with the same sequencing depth, which raised sequencing costs and weakened its ability to detect and accurately quantify high-quality circRNAs". There are multiple issues with this statement. In terms of the precision of circular RNA quantitation (e.g. as evaluated based on comparison among nanopore replicates as well as comparison to short-read RNA-seq data), the metrics presented for circFL-seq are in fact quite comparable to the metrics presented in the isoCirc paper, so there is no evidence that circFL-seq provides a better quantitation of circular RNAs. Moreover, given that the ground-truth is not known, an alternative interpretation to this observation is that circFL-seq may be biased towards highly expressed circular RNAs, and may lack the ability to discover moderately and lowly expressed circular RNAs. Overall, in comparing different methods, the authors should aim to provide an impartial discussion about the strengths and weaknesses of individual methods, and avoid over-interpreting the data in favor of their own method.

Because circFL-seq is more sensitive and more precise for BSJ detection, circFL-seq could also provide better quantitation. However, due to the advantage of quantification performance is not obvious, we have removed the advantage descriptions of circFL-seq for circRNA quantification.

4. The discussion about CDR1as is interesting. Can CIRI-long detect this circular RNA?

Thank you for your suggestion. Because CIRI-long only published mouse brain data, we employed the computational pipeline of CIRI-long on our circFL-seq data of HEK293 cell line. CIRI-long also successfully identified *CDR1as*. We also employed the computational pipeline of circFL-seq on isoCirc data but failed to detect *CDR1as*. Thus, we speculated that isoCirc data lack the sequenced reads of *CDR1as*, the reason of which could due to the experimental protocol. Because some circRNAs, including *CDR1as*, are sensitivity to RNase R (*Linda Szabo*1 *and Julia Salzman*, 2016, Nature Reviews Genetics), we hypothesized that digestion of linear transcripts by RNase R might require to be optimized in the protocol of isoCirc. We have removed the discussions about *CDR1as* in Discussion section.

5. The authors should also cite the BioRxiv preprint by Rahimi et al., (https://doi.org/10.1101/567164), which is another method for nanopore sequencing of circular RNAs.

We have cited the preprint paper in the introduction section as ref 13.

6. Overall, the manuscript is easy to read and follow, but it could benefit from a thorough editing by a language editor.

We have made a language editing by American Journal Experts (AJE).

Reviewer #2 (Recommendations for the authors):Major concerns:1. In comparison to isoCirc, circFL-seq identified fewer circRNA isoforms with higher read coverage of the detected circRNAs. This raises a concern that the outcome may result from that RCRT captures circRNA less efficiently than RCA, resulting in fewer circRNA molecules are captured in circFL-seq. The higher read coverage may simply come from sequencing the same circRNA molecule from the PCR amplification artifacts. This may also explain why circFL-seq cannot detect circRNAs with low read count or lowly expressed circRNAs. In this case, the authors cannot use back splice junction (BSJ) detection saturation as an indicator to compare the required read-depth between isoCirc and circFL-seq. Also, given the concern above, the "high read coverage" does not necessarily mean "high quality" nor "high accuracy" as claimed by the authors. The authors should address this concern before claiming on the benefits of high read coverage.

Thank you for the suggestion. We agreed that saturation curve was not appropriate and thus deleted related data from our revised manuscript (marked as Q2.1).

Comprehensive comparison between RCA and RCRT has been discussed above. And the worry about bias of high expressed circRNA has also been discussed above.

2. The advantages of circFL-seq over other existing technologies are not well-supported. For example, the authors claim that RCRT has lower residual linear RNA contamination than RCA, but the authors do not provide any data or evidence supporting the claim. Also, the authors claim that the circFL-seq gives higher circRNA read coverage; hence it is beneficial for circRNA quantification. However, the real "benefits" over other technologies (RNA-seq and isoCirc) for circRNA quantification are not clear since the RNA-seq (and isoCirc) quantification is significantly correlated with circFL-seq as demonstrated by the authors.

Sorry for the misunderstanding. We have performed the three-way comparison.

3. In the manuscript, the results are often comparable with known database and existing technologies when the authors focus on the "high quality" circRNAs only (circFL-seq read counts >= 5) which also have high expression level. The fact suggests that circFL-seq result is trust-worthy on "high quality" only. It also suggests that circFL-seq may fail to detect the lowly expressed circRNAs. The authors should note and discuss these limitations.

Sorry for confusing. The term “high quality” we used here was not equivalent to high expression level. With higher sequencing depth, lowly expressed circRNAs would become “high quality”. For lowly expressed circRNAs, circFL-seq was not more biased than isoCirc (Author response table 2).

General comments:1. The Figures and Supplemental Figures were not labeled with figure numbers, making it extremely difficult to read (especially some figures span across more than one page). The authors should label the figure number more clearly.2. Some figures are not clearly labeled. For example, in Figure 2j, what does each lane represent? Also, in Figure 3e, it is not clear what the y-axis is. The authors state that it is read coverage, but how come the value is from 0 to 1? In Figure 3d, the authors state that it is the correlation between HeLa and SKOV3, but it was not clear what axis is HeLa and SKOV3, respectively. Again, what does the shaded area mean in these figures (figure 3c, 3d, 3f, and 3g)? The authors should check through the figure label more carefully and correct them accordingly.

Sorry for the inconvenience. We have revised and labeled all figures and supplemental figures.

Specific comments:1. In the Abstract section, the authors claim that "… the detection of cancer-related fusion circRNAs…". However, the authors did not provide any data or literature suggesting that the fusion circRNAs they identified by circFL-seq are really "cancer-related" or biologically meaningful. The claim needs to be revised or proved.

We have removed the phrase “cancer-related” in our revised manuscript.

2. In Figure2—figure supplement 1a, the full-length circRNA reads contribute very little percentage of the total clean reads (~2-5%). How come the RCRT method generate so little full-length circRNA reads? The authors should comment on this and discuss it.

In our manuscript, we have shown the percentage of BSJs reads from RNA-seq is only 0.15%–0.32%, an amount ten times lower than that of full-length circRNA reads with circFL-seq. In addition, the percentage isoCirc ranged between 3.5% and 4.0%, which is comparable with circFL-seq.

3.In Figure 3—figure supplement 3, the author claim that more reads are required for ONT to confidently identify circRNAs. Doesn't this compromise the cost-efficient claim of circFL-seq made by the authors earlier? The authors should comment on this and discuss it.

Sorry for the misunderstanding. circFL-seq is developed for full-length circRNA sequencing. In the detection of BSJs, the cost of circFL-seq is much higher than RNA-seq. For the cost-efficient question in the full-length level, we want to compare circFL-seq to isoCirc. Due to the high noise of ONT reads, both circFL-seq and isoCirc have shown circRNAs with more read counts is more reliable. Because circFL-seq have the advantage to produce more reads. Thus, circFL-seq is a cost-efficient method for full-length circRNA sequencing.

4. In Figure 3—figure supplement 6a-d, how the saturation curves are calculated? It seems like isoCirc has much lower sequencing depth (~0.3 M) than circFL-seq. How do the authors compare the BSJ saturation in this case?

We have revised our discussion about the figure.

5. When comparing the validity of circFL-seq and isoCirc, why do the authors focus on top 100 expressed BSJ only? A better comparison should be the total BSJ in circFL-seq and isoCirc.

In this section, we want to find evidence of false discoveries from cDNA ligation of isoCirc. We focused on the false discoveries of top expressed circRNAs to eliminate the effect of high noise of nanopore reads and disturbance of low expressed circRNAs.

6. The authors should not use the same absolute circFL-seq BSJ read counts to define high-quality BSJs in isoCirc for the following reasons: (i) the read counts >= 5 in circFL-seq is arbitrary, there is no evidence suggesting that the isoCirc should use the same read counts to define high-quality BSJs. (ii) Since isoCirc captures more circRNAs, a lower BSJ read counts per circRNA is expected given the same sequencing depth. In both cases, lower BSJ read counts in isoCirc does not necessarily mean the BSJ is not "high-quality". Thus, the authors should not use absolute circFL-seq BSJ read counts as an indicator for the BSJ quality in isoCirc.

Sorry for confusing. We agreed that an absolute cutoff value was not fair for methods comparison. Actually, we collected all full-length circRNA (read counts >= 1) for comparison. The details of updated results were described above.

7. In the PLOD2 circFL-seq and RNA-seq example shown by the authors, the authors suggest that the circPLOD2 has lower exon skipping event than its parent linear RNA in HeLa cells. However, given that the BSJ is exactly the same between exon-skipped and non-exon-skipped circPLOD2, how does the back-splicing mechanism distinguish different parent linear RNA isoforms that selectively back-splices the non-exon-skipped linear RNA to generate specific circPLOD2 isoform in HeLa cells?

Sorry for the misunderstanding. We actually compared the exon skipping between two circular isoforms instead of between circRNA and its linear cognate.

8. Are the f-circ detected by circFL-seq generated by RNA fusion or genomic fusion? Although the RNA-seq suggests a genomic fusion, it does not completely eliminate the possibility of a genomic fusion. A genomic PCR followed by Sanger sequencing should be performed to validate the fusion junction of the genome.

Thank you for your suggestion. Details was described above.

Reviewer #3 (Recommendations for the authors):Considering that there are two recently published circRNA reconstruction tools based on nanopore sequencing, the authors should comprehensively compare their method with these two tools, and carefully discuss the advantages and disadvantages of these methods.Specific comments:1. In the Discussion section (line 260-262), the authors compared circFL-seq with the recently published CIRI-long method. Both circFL-seq and CIRI-long use a similar rolling circle reverse transcription strategy to amplify circRNAs. The authors may discuss the difference and (dis)advantages between their method and previous methods (isoCirc and CIRI-long).

We have discussed the differences of three methods and the limitation of circFL-seq.

2. In section "Comparison with RNA-seq and isoCirc for circRNA detection", the authors compared circFL-seq with the isoCirc method and found that circFL-seq produced more circular reads but identified fewer circRNA isoforms, which is an interesting result. Does it mean that isoCirc has a better sensitivity or higher false discovery rate in detecting lowly expressed circRNAs? The authors should include more comparison (e.g. venn diagram between three sets under different BSJ thresholds) between circFL-seq, isoCirc, and public circRNA database (e.g. circAtlas [PMID: 32345360, PMID: 30893614]) to demonstrate the advantages of their method.

A more detailed comparison was conducted.

3. The authors found six f-circ derived from GBF1-MACROD2 fusion and validated their junctions using Sanger sequencing. Besides, the authors also used short-read RNA-seq to validate the linear fusion junctions. What's the ratio of linear and circular transcript derived from these gene fusion loci? Is there any possibility that these f-circRNAs are derived from trans-splicing events? Considering that short-read RNA-seq data cannot effectively distinguish circular and linear transcripts, the authors may try to search for nanopore reads spanning the fusion region, which can provide direct evidence for these gene fusion events.

We have discussed the question in above.

4. Because of the high error rate of nanopore sequencing, the authors should compare the error rate of CS sequence before and after cRG correction to elucidate the ability to correct sequencing errors with the cRG mode.

We have discussed the question above.

5. The authors trained a random forest classifier to predict the strand origin of circular reads. How many CCRs were used as the training set, and how's the performance of the random forest classifier? The authors should provide more data about this step.

We have discussed the question above.

6. In section "Evaluation of quantification of full-length circRNAs", it would be nice if the authors could compare the quantification results between their method on nanopore reads and previous method (e.g. CIRIquant) on short-read RNA-seq data.

Thank you for suggestion. In the section, we have compared the quantification results between circFL-seq and RNA-seq (quantified by CIRI2), including Pearson correlation of expression level and DEC detection, which have shown comparable ability of BSJs quantification.

7. The authors used different names (circFL-seq, circfull) to denote their sequencing and data analysis methods. It would be better if they can unify the name, say, circFL-seq, which may avoid misunderstanding.

Thank you for suggestion. We have unified the name to circFL-seq in our manuscript.